# Time-Independent Information-Theoretic Generalization Bounds for SGLD

**Futoshi Futami***
Osaka University / RIKEN AIP
futami.futoshi.es@osaka-u.ac.jp

**Masahiro Fujisawa***†
RIKEN AIP
masahiro.fujisawa@riken.jp

## Abstract

We provide novel information-theoretic generalization bounds for stochastic gradient Langevin dynamics (SGLD) under the assumptions of smoothness and dissipativity, which are widely used in sampling and non-convex optimization studies. Our bounds are time-independent and decay to zero as the sample size increases, regardless of the number of iterations and whether the step size is fixed. Unlike previous studies, we derive the generalization error bounds by focusing on the time evolution of the Kullback–Leibler divergence, which is related to the stability of datasets and is the upper bound of the mutual information between output parameters and an input dataset. Additionally, we establish the first information-theoretic generalization bound when the training and test loss are the same by showing that a loss function of SGLD is sub-exponential. This bound is also time-independent and removes the problematic step size dependence in existing work, leading to an improved excess risk bound by combining our analysis with the existing non-convex optimization error bounds.

## 1 Introduction

Stochastic optimization, including stochastic gradient descent (SGD), is central to realizing practical large-scale or deep-learning models. There are currently considerable active discussions on accurately determining the generalization performance of models trained by SGD or its variants. In particular, stochastic gradient Langevin dynamics (SGLD) [13, 39, 29], a noisy variant of SGD, has garnered much attention in this type of study since it provides a useful theoretical framework for generalization error analysis based on the Langevin diffusion context [29]. Our study aims to contribute to a more accurate understanding and evaluation of the generalization performance for SGLD.

There are two main approaches to generalization analysis in SGLD. One is the *information-theoretic analysis* proposed by Russo and Zou [30] and Xu and Raginsky [40], by which a generalization error bound is derived using the mutual information (MI) between the learned parameters and the training dataset. Recently, some extensions using gradient information have been made to investigate the generalization properties of SGLD, for example, upper-bounding the MI with the norm of gradients [28] and the sum of gradient variances [25, 35, 37, 36]. Information-theoretic generalization bounds are applicable to a wide range of noisy iterative algorithms such as differentially private SGD [11] and stochastic gradient Hamiltonian Monte Carlo [5] modified to include a noisy momentum.

The other approach is *stability analysis*, by which the effects of changes in the learning algorithm due to the addition or removal of a single training data point on the generalization performance are investigated. Raginsky et al. [29] derived the non-asymptotic generalization and excess risk bound of SGLD via the exponential ergodicity of Langevin diffusion. Starting with the study by Raginsky

---

*Equal contribution.    †Corresponding author.

37th Conference on Neural Information Processing Systems (NeurIPS 2023).

et al. [29], there have been many attempts to improve the generalization analysis in SGLD from the stability perspective, such as those by Zhang et al. [43], Mou et al. [23] and Li et al. [20].

Unfortunately, these existing generalization bounds are *time-dependent*; namely, they diverge with increasing number of iterations unless the step size is adjusted so that the order of bound values is $\mathcal{O}(n^{-1})$ or $\mathcal{O}(n^{-1/2})$, where $n$ is the sample size (see Section 5 for details). Farghly and Rebeschini [10] attempted to avoid this problem by Wasserstein stability analysis through reflection coupling [9] under the smoothness and dissipativity [14] assumptions commonly used in sampling and non-convex optimization communities [29, 44]. Although their bounds bypass the divergence problem when taking a supremum over time, the geometry introduced for the reflection coupling yields an unnatural dependence on step size, resulting in a vacuous bound as the step size decreases (see Table 1).

In this paper, we provide novel generalization bounds for SGLD under smooth and dissipative loss functions obtained by the information-theoretic approach. We focus on the upper bound of the MI, namely, the Kullback–Leibler (KL) divergence between the distributions of parameters learned from different training datasets. We then analyze its time evolution caused by the update of the SGLD algorithm through the Fokker–Planck (FP) equation (Lemma 1). On the basis of this analysis, we obtain time-independent generalization error bounds that decay to zero as $n \to \infty$ regardless of the number of iterations or whether the step size is fixed (Theorem 4 and Corollary 1). Conventional information-theoretic generalization bounds [29, 28, 35, 36] are derived by bounding the MI between the parameters at all iterations and the training dataset. Therefore, these bounds grow linearly with the number of iterations, resulting in a time-dependent generalization error bound. Our analysis based on time evolution eliminates both this linearity issue and the unnatural dependence on step size (the inverse of step size) in the time-independent bound of Farghly and Rebeschini [10].

Another contribution is providing the first information-theoretic generalization bound and the excess risk bound when the same loss is used for training and the generalization performance evaluation. In the conventional information-theoretic approach, deriving generalization error bounds under this setting was challenging owing to the unknown tail behavior of a loss function of SGLD. We overcome this difficulty with our discovery that a smooth and dissipative loss function of SGLD is sub-exponential.

## 2 Preliminaries

### 2.1 Problem settings and stochastic gradient Langevin dynamics

We represent random variables in capital letters, such as $X$, and deterministic values in lowercase letters, such as $x$, and express the Euclidean inner product and distance as $\cdot$ and $\|\cdot\|$. Let $\mu$ be an unknown generating distribution on the instance space $\mathcal{Z}$ and $w \in \mathcal{W} \subseteq \mathbb{R}^d$ be the $d$-dimensional parameters such as weights of neural networks, where $\mathcal{W}$ is the space of the parameters. We consider a loss function $l : \mathcal{W} \times \mathcal{Z} \to \mathbb{R}$ and the following optimization problem:

$$\min_{w \in \mathcal{W}} L_\mu(w) := \mathbb{E}_Z[l(w, Z)] = \int_{\mathcal{Z}} l(w, Z)\mathrm{d}\mu(z),$$

which cannot be computed since $\mu$ is unknown. Instead, we typically minimize the empirical risk estimated using the dataset $S := \{Z_i\}_{i=1}^n$:

$$\min_{w \in \mathcal{W}} L_S(w) := \frac{1}{n} \sum_{i=1}^n l(w, Z_i),$$

where $\{Z_i\}_{i=1}^n$ are independent and identically distributed (i.i.d.) samples from $\mu$, i.e., $Z_i \overset{\text{i.i.d.}}{\sim} \mu$.

**Stochastic gradient Langevin dynamics.** In this paper, we use the SGLD algorithm [39] to solve the empirical risk minimization. SGLD utilizes the gradient information of the loss function; however, some loss functions, such as the 0-1 loss, are not differentiable. In this case, it is common to use the differentiable surrogate loss function $f : \mathcal{W} \times \mathcal{Z} \to \mathbb{R}$ (e.g., the cross-entropy loss) and minimize the following empirical risk: $F_S(w) := \frac{1}{n} \sum_{i=1}^n f(w, Z_i)$. Given a mini-batch $B \subset [n] := \{1, \cdots, n\}$ with $k = |B| \leq n$, we define its mini-batch version as

$$F(w, B) := \frac{1}{k} \sum_{i \in B} f(w, Z_i).$$

The SGLD algorithm updates the parameters using the following recursion:

$$W_{t+1} = W_t - \eta_t \nabla F(W_t, B_t) + \sqrt{2\beta_t^{-1}\eta_t}\xi_t, \quad W_0 \sim P_{W_0},$$

where $P_{W_0}$ is a given initial distribution, $\nabla F(w, B)$ is a stochastic gradient, $t$ is the number of iterations, $\eta_t$ is the step size, $\beta_t$ is the inverse temperature, and $(B_t)_{t=0}^{\infty}$ is an i.i.d. sequence of random variables distributed uniformly on $\{B \subset [n] : |B| = k\}$. In addition, $(\xi_t)_{t=0}^{\infty}$ is an i.i.d. sequence of standard Gaussian random variables, i.e., $\xi_t \sim \mathcal{N}(0, \mathbf{I}_d)$, where $\mathbf{I}_d$ is the $d$-dimensional identity matrix. The output parameters $W \in \mathcal{W}$ obtained using SGLD can be seen as the samples from a conditional distribution $P_{W|S} : \mathcal{Z}^n \to \mathcal{W}$. We express the $t$-th output of the SGLD algorithm as $W_t$.

## 2.2 Expected generalization error and its bounds

The focus of this paper is the *expected generalization error*, defined as

$$\text{gen}(\mu, P_{W|S}; L) \coloneqq \mathbb{E}_{S,W}[L_\mu(W) - L_S(W)], \tag{1}$$

where the expectation is taken over the joint distribution of $(S, W)$, i.e., $\mu^n \otimes P_{W|S}$.

**Information-theoretic generalization bounds.** Russo and Zou [30] and Xu and Raginsky [40] have shown that Eq. (1) can be bounded by the MI between the input dataset $S$ and the output parameters $W$ under the following sub-Gaussian assumption.

**Assumption 1** (sub-Gaussian losses). *A loss function $l(w, Z)$ is sub-Gaussian under $Z \sim \mu$ for all $w \in \mathcal{W}$, that is, there is a positive constant $\sigma_g^2$ such that $\log \mathbb{E}_Z[\exp(\lambda(l(w, Z) - \mathbb{E}l(w, Z)))] \leq \lambda^2 \sigma_g^2/2$ for all constant $\lambda \in \mathbb{R}$.*

For example, bounded or Lipschitz-continuous loss functions satisfy this assumption. Assumptions regarding the tail behavior of the loss function distributions as in the above are necessary for the information-theoretic generalization error analysis. Bu et al. [3] have investigated information-theoretic generalization bounds with another tail-behavior assumption such as sub-exponential losses.

We introduce the following standard information-theoretic generalization bound.

**Theorem 1** (Russo and Zou [30] and Xu and Raginsky [40]). *Suppose that Assumption 1 holds. Then, we have*

$$|\text{gen}(\mu, P_{W|S}; L)| \leq \sqrt{\frac{2\sigma_g^2}{n}I(W; S)}, \tag{2}$$

*under a training dataset $S = \{Z_i\}_{i=1}^n$ and the algorithm's output $W$, where $I(W; S)$ is the MI between $W$ and $S$.*

In the SGLD context, $I(W, S)$ of Eq. (2) can be upper-bounded in a form that incorporates the gradient variance [28, 25, 35, 36]. Given the output of the $T$-th iterate of the SGLD algorithm, $W_T$, the following upper bound can be obtained.

**Theorem 2** (Modified bound of Pensia et al. [28]). *Let $f(\cdot, z)$ be an $L$-Lipschitz continuous function, namely, there is a constant $L > 0$ such that $\|f(w, z) - f(\bar{w}, z)\| \leq L\|w - \bar{w}\|$ holds for all $w, \bar{w} \in \mathcal{W}$ and all $z \in \mathcal{Z}$. Then, we obtain*

$$I(W_T; S) \overset{\text{(i)}}{\leq} \sum_{t=0}^{T} \frac{d}{2} \log\left(1 + \frac{\beta_t \eta_t}{d}\text{Var}[\nabla f(W_t, B_t)|W_t]\right) \leq \sum_{t=0}^{T} \frac{d}{2}\log\left(1 + \frac{\beta_t \eta_t L^2}{d}\right), \tag{3}$$

*where $\text{Var}[\nabla f(W, B)|W] \coloneqq \mathbb{E}\mathbb{E}_B[\|\nabla_W f(W, B) - \mathbb{E}_B[\nabla_W f(W, B)]\|^2|W]$ is the conditional variance.*

Note that bound (i) can be obtained by the data-processing inequality [6].

The merit of such information-theoretic generalization bounds is that we can evaluate the bound value using the empirically estimated gradient variance per iteration. However, unfortunately, from Eq. (3), this bound is *time-dependent*; namely, the bound value can diverge unless the gradient variance or $\beta_t \eta_t$ approaches 0 as $T \to \infty$. This is due to the data-processing inequality when deriving upper bound (i) in Eq. (3). By the data-processing inequality, we obtain $I(W_T; S) \leq I(W^{(T)}; S)$, where

$W^{(T)} := (W_0, W_1, \cdots, W_T)$ denotes the joint random variables appearing in all the iterations in the algorithm. Since $W^{(T)}$ is treated simultaneously, the bound is inevitably linear in $T$.

Another limitation of the information-theoretic approach appears in the setting where training losses ($f$) are also used for performance evaluation, which is often employed in sampling and non-convex optimization studies of SGLD [29, 41]. In this setting, the generalization error is defined as

$$\text{gen}(\mu, P_{W|S}; F) := \mathbb{E}_{S,W}[F_\mu(W) - F_S(W)], \tag{4}$$

where $F_\mu := \mathbb{E}_Z[f(w, Z)]$. We cannot conduct the information-theoretic analysis for Eq. (4) because the tail behavior of the distribution of the training loss is unclear.

**Time-independent generalization bounds for Eq.** (4). To solve the above problems, Farghly and Rebeschini [10] provided the generalization error bounds of Eq. (4) from the stability perspective under the following assumptions widely used in the non-convex optimization analysis of SGLD [29, 41, 19].

**Assumption 2** (Smoothness). *For each $z \in \mathcal{Z}$, $f(\cdot, z)$ is differentiable and $M$-smooth. That is, there is a positive constant $M$ for all $w$, $\bar{w} \in \mathcal{W}$ and all $z \in \mathcal{Z}$ such that*

$$\|\nabla f(w, z) - \nabla f(\bar{w}, z)\| \leq M\|w - \bar{w}\|.$$

**Assumption 3** (Dissipativity [14]). *For each $z \in \mathcal{Z}$, $f(\cdot, z)$ is $(m, b)$-dissipative.* [1] *That is, there are positive constants $m$ and $b$ for all $w \in \mathcal{W}$ and $z \in \mathcal{Z}$ such that*

$$m\|w\|^2 - b \leq \nabla f(w, z) \cdot w.$$

The discussion regarding loss functions that satisfy Assumption 3 is presented in Appendix B.

Hereafter, we eliminate the time dependence of the step size and temperature by setting $\eta_t = \eta$ and $\beta_t = \beta$. With this notation, Farghly and Rebeschini [10] derived the following generalization bound.

**Theorem 3** (Farghly and Rebeschini [10]). *Suppose that Assumptions 2 and 3 hold. Assume that the initial law of $W_0$ has a finite fourth moment $\sigma$. Then, if $\eta \leq 1/2m$, for any $T \in \mathbb{N}$, we have*

$$|\text{gen}(\mu, P_{W_T|S}; F)| < C_1 \left( \eta T \wedge \frac{n(C_2 + 1)}{n - k} \right) \left( \frac{k}{n\eta^{1/2}} + \eta^{1/2} \right), \tag{5}$$

*where $(x \wedge y) = \min\{x, y\}$, and $C_1$ and $C_2$ are the positive constant terms w.r.t. $\{M, m, b, d, \beta, \sigma\}$ and $\{M, m, b, d, \beta\}$, respectively.*

Farghly and Rebeschini [10] utilized the Wasserstein stability on the basis of the contraction property of Langevin diffusion under reflection coupling. The important technique to derive the above bound is that we only focus on $W_T$ differently from $W^{(T)}$ of the information-theoretic approach when deriving the contraction property. In this way, the resulting bounds do not suffer from divergence as $T \to \infty$; however, it still has a problem. That is, Eq. (5) depends on the factor $\eta^{-1/2}$, which implies that it becomes vacuous or even diverges with decreasing $\eta (= \eta_T)$ as $T \to \infty$.

In this paper, we propose new generalization bounds to address the drawbacks of the information-theoretic and stability-based approaches. Specifically, the proposed bounds are established on the basis of the two expected generalization errors outlined in Eqs. (1) and (4), which remain time-independent and do not diverge as the step size decreases.

## 3 Time-independent generalization error bound for SGLD

Here, we explain our time-independent bound of $\text{gen}(\mu, P_{W|S}; L)$ for SGLD. We first introduce the main result (Section 3.1) and then summarize its proof outline (Sections 3.2 and 3.3). Finally, in Section 3.4, we provide a detailed discussion on our bound with concrete examples.

---

[1] This assumption holds not only for (strongly) convex losses but also for many practically used non-convex loss functions [24]. For example, it applies to non-convex loss functions with $l_2$ constraints and likelihood functions that satisfy Poincaré inequality [2, 32].

## 3.1 Main result

Our key idea is to derive the generalized error bound using the FP equation. To use the FP equation, we impose the following regularity condition for $P_{W_0}$.

**Assumption 4** (Regularity of the initial distribution). *The initial distribution of $W_0$: $P_{W_0}$ is a Gaussian distribution [2] with a finite variance $s^2 > 0$, which is independent of $\eta$ and $T$.*

Our analysis is also grounded in the time evolution of the FP equation using the logarithmic Sobolev inequality (LSI) [2] associated with $\pi$ described as follows. We state that $\pi$ satisfies the LSI with constant $c_{LS}$, if for any $\rho \ll \pi$, the following relation holds:

$$\mathrm{KL}(\rho|\pi) \leq c_{LS}\mathbb{E}\|\nabla \log \rho - \nabla \log \pi\|^2.$$

Raginsky et al. [29] showed the existence of $c_{LS}$ under Assumptions 2, 3, and $\beta \geq 2/m$. Note that $c_{LS}$ is expressed by the problem-dependent constant (see Appendix F.1 for details).

We now introduce our generalization error bound.

**Theorem 4.** *Suppose that Assumptions 1, 2, 3, and 4 are satisfied. Then, for any $\beta \geq 2/m$ and $\eta \in (0, 1 \wedge \frac{m}{5M^2} \wedge 4\beta c_{LS})$ and any $T \in \mathbb{N}$, we have*

$$|\mathrm{gen}(\mu, P_{W_T|S}; L)| \leq \sqrt{\frac{2c_1\sigma_g^2}{n}\left(1 \wedge \frac{\eta T}{4\beta c_{LS}}\right)(V_\nabla + c_2)}, \tag{6}$$

*where $c_1$, $c_2$, and $V_\nabla$ are the positive constant terms w.r.t. $\{M, m, b, d, \beta, s^2\}$.*

The above theorem shows $|\mathrm{gen}(\mu, P_{W_T|S}; L)| = \mathcal{O}(\sqrt{(\eta T \wedge 1)/n})$, which implies time independence since it does not diverge even if $T \to \infty$ and thus converges as $n \to \infty$.

In Eq. (6), the term $V_\nabla$ corresponds to *stability*, which is expressed as the upper bound of the difference of the expected conditional gradients with respect to changes in training datasets at each iteration. This shows a certain similarity to existing information-theoretic generalization bounds, such as Theorem 2, expressed by the variance of gradients with respect to the training datasets. This similarity is discussed in detail in Section 3.4. Additionally, detailed information on the explicit expression of $c_1$, $c_2$, and $V_\nabla$ can be found in Appendix F.

## 3.2 Proof outline of Theorem 4

In this section, we present how to derive our bound in Theorem 4. Our aim here is to share the ideas behind our analysis and an outline of the proof, providing the detailed proof in Appendix F.

We adopt the information-theoretic approach and focus on the MI in Eq. (2). By using the Jensen inequality, we have the following upper bound of the MI:

$$I(W_T; S) \leq \mathbb{E}_{S,S'}\mathrm{KL}(P_{W_T|S}|P_{W_T|S'}), \tag{7}$$

where $S$ and $S'$ are random variables drawn independently from $\mu^n$, and $\mathrm{KL}(P_{W_T|S}|P_{W_T|S'})$ is the KL divergence from $P_{W_T|S'}$ to $P_{W_T|S}$. Note that this KL divergence indicates the stability of the learned parameter from two datasets, $S$ and $S'$. We also note that $P_{W_T|S'}$ can be regarded as the data-dependent prior. Thus, this KL divergence is tighter than that of the data-independent prior, which is often used in the probably approximately correct (PAC)-Bayes bound [3].

The key idea is to analyze *the time evolution of the KL divergence*, which is summarized in the following lemma:

**Lemma 1.** *Suppose that the same assumptions in Theorem 4 hold. Then, for any $t \in \mathbb{N}$, we have*

$$\mathrm{KL}(P_{W_t|S}|P_{W_t|S'}) \leq e^{-\frac{\eta}{4\beta c_{LS}}}\mathrm{KL}(P_{W_{t-1}|S}|P_{W_{t-1}|S'}) + \eta V_\Delta + c_3\eta, \tag{8}$$

*where $V_\Delta$ and $c_3$ is the constant term w.r.t. $\{M, m, b, d, \beta, s^2\}$.*

---

[2] The Gaussian assumption can be relaxed, e.g., to a Gaussian mixture, in the theorems and corollaries shown in this paper. The detailed discussions are provided in Appendix F.3.   [3] We can confirm this from the fact that $\mathbb{E}_{S,S'}\mathrm{KL}(P_{W_T|S}|P_{W_T|S'}) = \mathbb{E}_S\mathrm{KL}(P_{W_T|S}|P_{W_T}) - \mathbb{E}_{S'}\mathrm{KL}(P_{W_T|S'}|P_{W_T})$, where $P_{W_T}$ is a data-independent prior distribution.

We will discuss the details of Lemma 1 in Section 3.3. By recursively applying Eq. (8) from $t = 0$ to $T$, we obtain

$$\mathrm{KL}(P_{W_T|S}|P_{W_T|S'}) \leq \frac{1 - e^{-\frac{\eta T}{4\beta c_{\mathrm{LS}}}}}{1 - e^{-\frac{\eta}{4\beta c_{\mathrm{LS}}}}} \eta \left(V_\nabla + c_3\right) \overset{(\mathrm{i})}{\leq} 4\beta c_{\mathrm{LS}} \left(1 \wedge \frac{\eta T}{4\beta c_{\mathrm{LS}}}\right) \frac{1}{1 - \frac{\eta}{4\beta c_{\mathrm{LS}}}} \left(V_\nabla + c_3\right),$$

which is based on the fact that $\mathrm{KL}(P_{W_0|S}|P_{W_0|S'}) = 0$. Note that bound (i) can be obtained from $e^{-\frac{\eta}{4\beta c_{\mathrm{LS}}}} < 1 - \frac{\eta}{4\beta c_{\mathrm{LS}}} + \frac{\eta^2}{16\beta^2 c_{\mathrm{LS}}^2}$ when $\frac{\eta}{4\beta c_{\mathrm{LS}}} \leq 1$, and $e^{-\frac{\eta T}{4\beta c_{\mathrm{LS}}}} \geq 1 - \frac{\eta T}{4\beta c_{\mathrm{LS}}}$.

### 3.3  Proof outline of Lemma 1 under the continuous Langevin diffusion

Here, we organize our ideas for the proof of Lemma 1 that are important in the derivation of Theorem 4. For simplicity, we now provide an intuitive explanation and an outline of the proof under the continuous Langevin diffusion setting. Note that the results of Theorem 4 and Lemma 1 are based on the SGLD setting, and their proofs are shown in Appendix F.

The Langevin diffusion is defined as

$$\mathrm{d}W_t = -\nabla F(W_t, S)\mathrm{d}t + \sqrt{2\beta^{-1}}\mathrm{d}H_t, \tag{9}$$

where $\mathrm{d}H_t$ is the standard Brownian motion in $\mathbb{R}^d$. Note that, in this context, $t$ expresses the continuous time and the *full-batch* gradient $\nabla F(W_t, S)$ is used. The stationary distribution of Eq. (9) is given as the Gibbs distribution $\pi(\mathrm{d}w) \propto \exp(-\beta F(w, S))$.

With some abuse of notation, let us denote $P_{W_t|S}$ as the conditional distribution obtained using Eq. (9) and express its density as $\rho_t$. Then, the FP equation [2] for Eq. (9) can be obtained as

$$\frac{\partial \rho_t(w, t)}{\partial t} = \nabla \cdot \left(\frac{1}{\beta}\nabla\rho_t + \rho_t \nabla F(w, S)\right). \tag{10}$$

Similarly, we can define the Langevin diffusion when we use dataset $S'$ and the conditional distribution using that diffusion as $P_{W_t|S'}$ with the density $\gamma_t$, obtaining the FP equation in the form of $\rho_t$ replaced by $\gamma_t$ in Eq. (10).

Now we analyze the time evolution of $\mathrm{KL}(P_{W_t|S}|P_{W_t|S'}) = \mathrm{KL}(\rho_t|\gamma_t)$ at time $t$, i.e., $\partial \mathrm{KL}(\rho_t|\gamma_t)/\partial t$. By utilizing the FP equations of $\rho_t$ and $\gamma_t$ and the Cauchy–Schwartz inequality, we obtain the following upper bound:

$$\frac{\partial \mathrm{KL}(\rho_t|\gamma_t)}{\partial t} \leq -\frac{1}{2\beta}\mathbb{E}\|\nabla\log\rho_t - \nabla\log\gamma_t\|^2 + \frac{\beta}{2}\mathbb{E}\|\nabla F(W_t, S) - \nabla F(W_t, S')\|^2. \tag{11}$$

The second term on the right-hand side of Eq. (11) represents the stability of the gradient with respect to the randomness of the training dataset $S, S' \sim \mu^n$, which leads to $V_\nabla$ in Lemma 1 under the SGLD setting. Hereafter, we define $\mathbb{E}\|\nabla F(W_t, S) - \nabla F(W_t, S')\|^2$ as $\widetilde{V}_{\nabla_t}$.

By introducing $\nabla\log\pi(w)$ into $\mathbb{E}\|\nabla\log\rho_t - \nabla\log\gamma_t\|^2$ in Eq. (11), we obtain

$$\frac{\partial \mathrm{KL}(\rho_t|\gamma_t)}{\partial t} \leq -\frac{1}{4\beta}\mathbb{E}\|\nabla\log\rho_t - \nabla\log\pi\|^2 + \frac{1}{2\beta}\mathbb{E}\|\nabla\log\pi\|^2 + \frac{1}{\beta}\mathbb{E}\nabla\log\rho_t \cdot \nabla\log\gamma_t + \frac{\beta}{2}\widetilde{V}_{\nabla_t}$$

$$\leq -\frac{1}{4\beta c_{\mathrm{LS}}}\mathrm{KL}(\rho_t|\pi) + \frac{1}{2\beta}\Omega(\rho_t, \gamma_t, \pi) + \frac{\beta}{2}\widetilde{V}_{\nabla_t}$$

$$\leq -\frac{1}{4\beta c_{\mathrm{LS}}}\left(\mathrm{KL}(\rho_t|\gamma_t) + \mathbb{E}\log\frac{\gamma_t}{\pi}\right) + \frac{1}{2\beta}\Omega(\rho_t, \gamma_t, \pi) + \frac{\beta}{2}\widetilde{V}_{\nabla_t}, \tag{12}$$

where the first inequality is from the fact that $-x^2 \leq -\|x - y\|^2/2 + y^2$ for $x, y \in \mathbb{R}^d$ and the second one is from the LSI. We introduced $\Omega(\rho_t, \gamma_t, \pi) := \mathbb{E}_{\rho_t}\|\nabla\log\pi\|^2 + 2\mathbb{E}_{\rho_t}\nabla\log\rho_t \cdot \nabla\log\gamma_t$ to simplify the notation.

By integrating $e^{\frac{t}{4\beta c_{\mathrm{LS}}}} \frac{\partial \mathrm{KL}(\rho_t|\gamma_t)}{\partial t}$ in Eq. (12) over $t \in [0, \eta]$ and rearranging it, we obtain

$$\mathrm{KL}(\rho_\eta|\gamma_\eta) \leq e^{\frac{-\eta}{4\beta c_{\mathrm{LS}}}}\mathrm{KL}(\rho_0|\gamma_0)$$

$$+ \int_0^\eta e^{\frac{-(\eta-t)}{4\beta c_{\mathrm{LS}}}}\left(\frac{\beta}{2}\widetilde{V}_{\nabla_t} - \frac{1}{4\beta c_{\mathrm{LS}}}\mathbb{E}\log\frac{\gamma_t}{\pi} + \frac{1}{2\beta}\Omega(\rho_t, \gamma_t, \pi)\right)\mathrm{d}t. \tag{13}$$

In Appendix F, we show that the terms related to $\pi$ in Eq. (13) can be bounded by using the techniques of Raginsky et al. [29] and Vempala and Wibisono [32].

We next derive an upper bound for the following terms in Eq. (13): $\mathbb{E}[\log\frac{\gamma_t}{\pi}]$ and $\Omega(\rho_t, \gamma_t, \pi)$ by using the *parametrix method* for the FP equation [12, 27], which allows us to expand the FP equation's solution via the heat kernel. On the basis of this expansion, we can upper bound Eq. (13) as

$$\int_0^\eta e^{\frac{-(\eta-t)}{4\beta c_{\mathrm{LS}}}} \left( -\frac{1}{4\beta c_{\mathrm{LS}}}\mathbb{E}\log\frac{\gamma_t}{\pi} + \frac{1}{2\beta}\Omega(\rho_t,\gamma_t,\pi) \right) \mathrm{d}t \le \mathcal{O}(\eta). \tag{14}$$

By combining Eq. (14) with Eq. (13), we obtain the continuous version of Lemma 1.

The same procedure can be used for the SGLD setup. The difference from the continuous Langevin diffusion case is that the discretization errors and the effects of using a stochastic gradient are taken into account, resulting in the appearance of an additional constant [4] in the above bounds (see Appendix F for details).

### 3.4 Additional discussion on our bound in terms of stability

We conclude this section by presenting further discussion on our bound in terms of stability with a concrete example.

As shown in Eq. (7), the information-theoretic generalization bound is closely related to the stability in KL divergence under the different training datasets. However, our bound in Theorem 4 incorporates the constant term $c_2$, which is irrelevant to stability, alongside the stability term $V_\nabla$. If we can avoid the occurrence of $c_2$, the resulting upper bound of $\mathrm{KL}(P_{W_t|S}|P_{W_t|S'})$ would be dominantly expressed by $V_\nabla$, and as a result, we may obtain a bound where the relationship between generalization and stability is more directly represented.

The problematic constant term $c_2$ arises from $c_3\eta$ in Lemma 1 analyzing the time evolution of stability in KL divergence. Specifically, the term $c_3\eta$ is the byproduct of treating the general dissipative function using LSI. Actually, it is possible to avoid the problematic constant term $c_3\eta$ and derive bounds that are evaluated solely on the basis of stability-related metrics in specific examples, such as strongly convex or bounded (non-convex) losses with $l_2$-regularization. For simplicity, we show this fact using the following theorem under the Langevin diffusion (LD) setting, where the probability induced by Eq. (9) is expressed as $P_{W_T|S}$.

**Theorem 5.** *Suppose that Assumptions 1 and 2 are satisfied and that $F(w,z)$ is R-strongly convex $(0 < R < \infty)$. Then, for any $T \in \mathbb{R}_+$, we have*

$$\frac{\partial \mathrm{KL}(\rho_t|\gamma_t)}{\partial t} \le -\frac{R}{4}\,\mathrm{KL}(\rho_t|\gamma_t) + \frac{\beta}{2}\mathbb{E}\|\nabla F(W_t,S) - \nabla F(W_t,S')\|^2, \tag{15}$$

*and*

$$|\mathrm{gen}(\mu, P_{W_T|S}; L)| \le \sqrt{\frac{2\beta\sigma_g^2}{n}\int_0^T e^{-\frac{(T-t)R}{4}}\mathbb{E}\|\nabla F(W_t,S) - \nabla F(W_t,S')\|^2\mathrm{d}t}. \tag{16}$$

A similar bound in Eq. (16) (with $R$ replaced by $\lambda/e^{8\beta C}$) can be obtained for bounded non-convex losses with $l_2$-regularization, where $F(w,z) = F_0(w,z) + \frac{\lambda}{2}\|w\|^2$ $(0 < \lambda < \infty)$ and $F_0(w,z)$ is $C$-bounded $(0 \le C < \infty)$. The full proof is summarized in Appendix F.4.

When comparing with Lemma 1, we can see that, in Eq. (15), stability-unrelated constants do not appear in the time evolution of KL divergence at each time step. Therefore, the resulting generalization bound is also independent of such constants. Furthermore, when compared with Theorem 2, which adds up the stability terms at all time steps, our bound is dominated by the stability terms near the final time step, as those at earlier time steps decrease geometrically by $e^{-\frac{R}{4}}$. This indicates that the stability around the initial time steps is of lesser importance in evaluating the final generalization performance.

Note that our bounds are closely related to the bound indicated in Proposition 9 of Mou et al. [23], which was also derived by focusing on stability. The bound of Mou et al. [23] primarily assesses

---

[4] This constant is evaluable (see Vempala and Wibisono [32] or Kinoshita and Suzuki [19]).

generalization errors focusing on the *gradient norm* near the conclusion of training. In contrast, our bounds evaluate it through the norm of *differences in gradients*, emphasizing the state in the proximity of training completion. In other words, our bound allows for the evaluation of generalization errors using a stability measure that is more closely related to generalization performance than the gradient norm. This benefit originates from our approach, which tracks the time evolution of MI-related stability in Eq. (15) on the basis of information-theoretic generalization bounds, in contrast to the PAC-Bayes bounds derived from the direct analysis of stability measures as in Mou et al. [23].

## 4 Generalization analysis for SGLD directly using a training loss

In this section, we consider the setting that the generalization performance is measured by a training loss $f$ directly as in Eq. (4). We show that this is possible by demonstrating that loss functions of SGLD are sub-exponential under smooth and dissipative assumptions (Section 4.1). On the basis of this fact, we obtain for the first time an information-theoretic generalization bound of SGLD that is similar to Theorem 4. Finally, combining these results with existing optimization error bounds provides an excess risk bound with improved convergence (Section 4.2).

### 4.1 Smooth and dissipative loss function of SGLD is sub-exponential

To perform an information-theoretic analysis for SGLD, it is necessary to know the tail behavior of $f(W, Z)$. Our contribution here is showing that a loss function of SGLD under smooth and dissipative assumptions is sub-exponential.

**Theorem 6.** *Suppose that Assumptions 2, 3 and 4 are satisfied. Let $P_{W_T} = \mathbb{E}_S[P_{W_T|S}]$ be the marginal distribution of the output obtained using the SGLD algorithm at the $T$-th iteration. Then, for any $\eta \in (0, 1 \wedge \frac{m}{5M^2})$ and $T \in \mathbb{N}$, $f(W_T, Z)$ is sub-exponential under the distribution $P_{W_T} \otimes \mu$. That is, there exist positive constants $\sigma_e^2$ and $\nu$ w.r.t. $\{m, \beta, M, b, d, s^2\}$ [5] such that*

$$\log \mathbb{E}_{W_T \otimes Z} \left[ e^{\lambda(f(W_T, Z) - \mathbb{E}_{W_T \otimes Z}[f(W_T, Z)])} \right] \leq \frac{\sigma_e^2 \lambda^2}{2} \quad for \ all \ |\lambda| < \frac{1}{\nu}.$$

*Proof sketch.* First, note that under Assumptions 2 and 3, for any $z \in \mathcal{Z}$, we obtain

$$\frac{m}{3}\|w\|^2 - \frac{b}{2}\log 3 \leq f(w, z) \leq \frac{M}{2}\|w\|^2 + M\sqrt{\frac{b}{m}}\|w\| + A, \tag{17}$$

where $A$ is a positive constant (see Lemma 8 in Appendix G.1 for its explicit form). By employing Lemma B.2 from Farghly and Rebeschini [10], we can show the following fact: for any $p \in \mathbb{N}$,

$$\mathbb{E}\|W_T\|_2^{2p} \leq \mathbb{E}\|W_0\|_2^{2p} + c(p), \tag{18}$$

where

$$c(p) := \frac{1}{m}\left(\frac{6}{m}\right)^{p-1}\left(1 + \frac{2^{2p}p(2p-1)d}{m\beta}\right)\left[\left(2b + 8\frac{M^2}{m^2}b\right)^p + 1 + 2\left(\frac{d}{\beta}\right)^{p-1}(2p-1)^p\right].$$

This implies that $W_T$ is a sub-Gausssian random variable according to Proposition 2.5.2 in Vershynin [33]. To show the sub-exponential property, we directly upper-bound $\mathbb{E}_{W_T \otimes Z}[e^{\lambda(f(W_T, Z) - \mathbb{E}_{W_T \otimes Z}[f(W_T, Z)])}]$ by considering the Taylor expansion of the exponential moment and using Eqs. (17) and (18) (see Appendix G.2 for the complete proof). □

**Remark 1.** *In previous information-theoretic analysis studies [28, 25, 35], it is often assumed that a loss function $l(w, Z)$ is sub-Gaussian under the distribution $\mu$ for all $w \in \mathcal{W}$. In contrast, Theorem 6 holds under the distribution $P_{W_T} \otimes \mu$, not conditioned on $w \in \mathcal{W}$.*

We can interpret the sub-exponential property of SGLD intuitively as follows. Under Assumptions 2 and 3, the loss function grows at most as a quadratic function shown in Eq. (17). The conditional distribution of the parameters follows the Gaussian distribution, and the square of the Gaussian random variable is known as the chi-square ($\chi^2$) random variable [34]. According to these facts, we expect that the behavior of the loss function resembles that of the $\chi^2$-random variable; therefore, it is sub-exponential since the $\chi^2$-distribution is also sub-exponential [34]. Theorem 6 validates this intuition.

---

[5] The explicit form of $\sigma_e^2$ and $\nu$ can be seen in Appendix G.2.

## 4.2 Generalization bounds for SGLD using the same loss for training and evaluation

On the basis of Theorem 6, we can derive the following information-theoretic generalization bound for SGLD even if a surrogate loss is not used. In contrast to Theorem 4, an assumption regarding the tail behavior of a loss function such as Assumption 1 is not necessary.

**Corollary 1.** *Suppose that Assumptions 2, 3, and 4 are satisfied. Then, for any $\beta \geq 2/m$, $\eta \in (0, 1 \wedge \frac{m}{5M^2} \wedge 4\beta c_{\mathrm{LS}})$, and $T \in \mathbb{N}$, we obtain*

$$|\mathrm{gen}(\mu, P_{W_T|S}; F)| \leq \Psi^{*-1}\left(\frac{c_1}{n}\left(1 \wedge \frac{\eta T}{4\beta c_{\mathrm{LS}}}\right)(V_\nabla + c_2)\right),$$

*where*

$$\Psi^{*-1}(y) = \begin{cases} \sqrt{2\sigma_e^2 y} & \text{if } y \leq \frac{\sigma_e^2}{2\nu} \\ \nu y + \frac{\sigma_e^2}{2\nu} & \text{otherwise} \end{cases},$$

*$c_1$ and $c_2$ are the same as in Theorem 4, and $\sigma_e^2$ and $\nu$ are the same as in Theorem 6.*

*Proof sketch.* This is the direct consequence of the sub-exponential property from Theorem 6 and the upper bound of MI in Eq. (7) (see Appendix G.3 for the complete proof). $\square$

**Remark 2.** *Despite the assumptions of Corollary 1 being the same as those made by Farghly and Rebeschini [10] except for the initial distribution and step size, the resulting bound becomes 0 as $n \to \infty$ without being dependent on inverse stepsize.*

We conclude this section by introducing our excess risk bound. Let us define the excess risk as follows: $\mathrm{Excess}(\mu, P_{W|S}) := \mathbb{E}_{W,S}[F_\mu(W) - F_\mu(w^*)]$, where $w^* = \mathrm{argmin}_{w \in \mathcal{W}} F_\mu(w)$. Under this definition, we derive the following upper bound for the excess risk by utilizing Corollary 1.

**Corollary 2.** *Suppose that Assumptions 2, 3, and 4 are satisfied. Then, for any $\beta \geq 2/m$, $\eta \in (0, 1 \wedge \frac{m}{5M^2} \wedge 4\beta c_{\mathrm{LS}})$, and $T \in \mathbb{N}$, we obtain*

$$\mathrm{Excess}(\mu, P_{W_T|S}) = \mathcal{O}\left(\sqrt{\frac{(\eta T \wedge 1)}{n}} + e^{-\eta T/c_{LS}} + \sqrt{\eta} + c_{\mathrm{err}}\right),$$

*where $c_{\mathrm{err}}$ is the positive constant w.r.t. $\{M, m, b, d, \beta\}$ corresponding to the optimization error.*

We show the complete proof in Appendix G.4. In contrast with the existing excess risk studies, our bound does not diverge with increasing $t$ owing to the time-independent generalization bound in Corollary 1.

## 5 Related studies and discussion

In this section, we compare our generalization bounds with those in related studies. Table 1 shows the order of each bound value along with its assumptions for a loss function.

**SGLD analysis with/without changing losses.** The existing generalization error bounds in Table 1 are time-dependent; namely, we need to impose restrictive conditions for the step size $\eta$ in terms of $t$ to achieve a generalization bound that decays to zero with increasing sample size [29, 28, 25, 35] (see the right column in Table 1). Some important applications of SGLD do not satisfy these conditions. For instance, the short-run Markov chain Monte Carlo [26] method used in energy-based models [17] adopts SGLD with a *fixed* step size. Another example is the cyclic SGLD [42] used in deep learning, where the step size is *periodically increased or decreased* to facilitate escape from local optima.

Farghly and Rebeschini [10] first analyzed the generalization error of SGLD by using smoothness and dissipative assumptions, which are broadly used in sampling and non-convex optimization studies [29, 41, 4, 44]. Their bound is time-independent; the bound does not diverge with time and achieves the order $\mathcal{O}(n^{-1/2})$. However, the bound depends on the inverse of step size $\eta^{-1/2}$ owing to the reflection coupling [9], which results in the unnatural behavior of decreasing $\eta$ with increasing $t$. Farghly and Rebeschini [10] also derived a bound that does not suffer from this problem by assuming the Lipschitz loss function with weight decay; however, these assumptions excessively restrict the

Table 1: Comparison of our bounds with those in existing studies. Our bounds are time-independent and bounded even if $\eta \to 0$. (**I**) denotes the information-theoretic approach and (**S**) denotes the stability analysis approach. The symbol * means that the sub-Gaussian assumption is unnecessary for our bounds when using the same loss for training and generalization performance evaluation. Namely, our bounds can be derived under more relaxed assumptions for a loss function in this case.

| Study | Assumptions for a loss function | Expected generalization error bound |
|---|---|---|
| (**S**) Raginsky et al. [29] (Thm. 2.1.) | Dissipative, Smoothness | $\mathcal{O}(\eta t + e^{-\eta t/c} + 1/n)$ |
| (**S**) Mou et al. [23] (Thm. 1.) | Bounded, Lipschitz | $\mathcal{O}(\sqrt{\eta t}/n)$ |
| (**S**) Mou et al. [23] (Thm. 2.) | Lipschitz, Sub-Gaussian, (Weight decay) [6] | $\mathcal{O}(\sqrt{\eta \log(t+1)/n})$ |
| (**I**) Pensia et al. [28] (Cor. 1.) | Lipschitz, Sub-Gaussian | $\mathcal{O}(\sqrt{\eta t/n})$ |
| (**I**) Negrea et al. [25] (Thm. 3.1.) | Sub-Gaussian | $\mathcal{O}(\sqrt{\eta t/n})$ |
| (**S**) Farghly and Rebeschini [10] (Thm. 3.1.) | Lipschitz, Smoothness, Weight decay | $\mathcal{O}((\eta t \wedge 1)(1/n + \sqrt{\eta}))$ |
| (**S**) Farghly and Rebeschini [10] (Thm. 4.1.) | Dissipative, Smoothness | $\mathcal{O}((\eta t \wedge 1)(\sqrt{\eta^{-1}}/n + \sqrt{\eta}))$ |
| (**I**) Wang et al. [35] (Thm. 1.) | Sub-Gaussian | $\mathcal{O}(\sqrt{\eta t/n})$ |
| (**I**) Ours (Thm. 4 and Cor. 1) | Dissipative, Smoothness, Sub-Gaussian* | $\mathcal{O}(\sqrt{(\eta t \wedge 1)/n})$ |

class of loss functions and algorithms. In contrast to these bounds, our bound is time-independent and does not require scaling $\eta$, $t$, and $n$ to achieve $\mathcal{O}(n^{-1/2})$.

**Time evolution analysis of MI via FP equation.** The analysis of SGLD using the FP equation has been successfully used in the convergence analysis of SGLD [32, 19]. These studies present analyses of the discretization errors and convergence properties of the unadjusted Langevin algorithm, SGLD, and variance-reduction SGLD (SVRG-LD) [8], comparing them with the continuous Langevin dynamics through the FP equation.

In generalization error analysis, the FP equation is mainly used to analyze the time evolution of KL divergence appearing in a generalization bound on the basis of the stability approach. Li et al. [20] analyzed the time evolution of the KL divergence between the probability densities of the parameters obtained from two training datasets that differ by only one data point under the bounded loss assumption. Mou et al. [23] also studied the KL divergence and Hellinger divergence, and they derived a generalization error bound on the basis of the PAC-Bayes notion [22]. Our idea is similar to these: we analyze the time evolution of the KL divergence between the probability densities of the parameters obtained from two training datasets. The differences between our approach and other approaches are twofold. First, we do not assume weight decay or Lipschitz continuity but instead derive our analysis assuming smoothness and dissipativity. Second, the KL divergence we analyzed is tighter than that of the PAC-Bayes bound with data-independent prior dealt by Mou et al. [23].

## 6 Limitations and future work

In this paper, we provide a generalization analysis of SGLD, where Gaussian noise is a fundamental assumption for our theoretical results. Thus, it is difficult to extend our analysis to other noisy iterative algorithm variants with a different noise, such as differentially private SGD with Laplace or uniform noise [36]. Another limitation of this study is that we have estimated the sub-exponential parameter roughly with respect to the dimensions of the model parameters. Further investigation of the sub-exponentiality of smooth and dissipative losses, and improvement of the dependence on dimensionality, are crucial for enhancing the practicality of our generalization bounds. The sub-exponential property of a loss function is expected to be helpful in fields other than generalized error analysis. For example, this property opens up room for new theoretical analysis policies that employ useful concentration and transport inequalities [34] in the sampling and optimization context. We hope that the analysis presented in this paper goes beyond generalization analysis and provides valuable insights into understanding the characteristics of machine learning.

---

[6] The order of the bound in Mou et al. [23] varies with the choice of regularization parameters and decay factors. In this paper, we adopt the order of this bound in Table 1 of Farghly and Rebeschini [10]. For a more comprehensive discussion, we refer to Section 5.2 of Mou et al. [23].

## Acknowledgments and Disclosure of Funding

We sincerely appreciate the anonymous reviewers for their insightful feedback. FF was supported by JSPS KAKENHI Grant Number JP23K16948. FF was supported by JST, PRESTO Grant Number JPMJPR22C8, Japan. MF was supported by RIKEN Special Postdoctoral Researcher Program. MF was supported by JST, ACT-X Grant Number JPMJAX210K, Japan.

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
