# A  Notation used in the main paper

We summarize the notation we used in the main part of this paper.

| Category | Symbol | Meaning |
|---|---|---|
| Scalars and constants | $n \in \mathbb{N}$ | The sample size |
| | $w \in \mathbb{R}$ | Model parameters (deterministic) |
| | $w^* \in \mathbb{R}$ | $\mathrm{argmin}_{w \in \mathcal{W}} L_\mu(w)$ (deterministic) |
| | $W \in \mathbb{R}$ | Model parameters (random variables) |
| | $t, T \in \mathbb{N}$ | An iteration of the SGLD algorithm |
| | $W^{(T)}$ | The joint random variables appearing in all the iterations until $T$ |
| | $d \in \mathbb{R}$ | The number of parameter dimensions |
| | $k \in \mathbb{N}$ | The number of samples in a mini-batch $B\ (\leq n)$ |
| | $\mathbf{I}_d$ | Identity matrix with $d$ rows and $d$ columns |
| | $\eta_t(=\eta) \in \mathbb{R}$ | The learning rate |
| | $\beta_t(=\beta) \in \mathbb{R}$ | The inverse temperature |
| | $\xi_t \in \mathbb{R}$ | Gaussian noise sampled from $\mathcal{N}(0, \mathbf{I}_d)$ |
| | $\sigma_g^2$ | A positive constant for sub-Gaussian random variables |
| | $L$ | A positive constant in the Lipschitz continuous function |
| | $M$ | A positive constant in the smoothness condition |
| | $m, b$ | Positive constants in the dissipative condition |
| | $s^2$ | A positive finite Gaussian variance for the initial parameter distribution $P_{W_0}$ |
| | $\sigma \in \mathbb{R}$ | A finite fourth moment of the initial parameters $W_0$ |
| | $c_{\mathrm{LS}} \in \mathbb{R}$ | The logarithmic Sobolev constant |
| | $\bar{V}_{\nabla_t} \in \mathbb{R}$ | The expected value of $\mathbb{E}\|\nabla F(W_t, S) - \nabla F(W_t, S')\|^2$ |
| | $\sigma_e^2$ | A positive constant for sub-exponential random variables |
| | $c_{\mathrm{err}} \in \mathbb{R}$ | The constant w.r.t. $\{M, m, b, d, \beta\}$ corresponding to the optimization error |
| Sets and sequences | $\mathcal{Z}$ | The instance space |
| | $\mathcal{W}$ | The parameter space |
| | $\mathbb{R}, \mathbb{R}_+$ | The set of real numbers and that of positive real numbers |
| | $\mathbb{N}$ | The set of natural numbers |
| | $[n] := \{1, \ldots, n\}$ | The set of all integers between $1$ and $n$ |
| | $S, S' := \{Z_i\}_{i=1}^n (\in \mathbb{R})$ | The i.i.d. samples from $\mu^n$ |
| | $B \subset [n]$ | A mini-batch set |
| | $(B_t)_{t=1}^\infty$ | An i.i.d. sequence of random variables specifying indexes |
| | $(\xi)_{t=1}^\infty$ | An i.i.d. sequence of Gaussian noise $\xi_t$ |
| Probability and information theory | $\mu$ | An unknown data generating distribution |
| | $P_{W\|S}$ | A conditional distribution w.r.t. $W$ given $S$ via SGLD (or the continuous Langevin diffusion) |
| | $\mathcal{N}(\mathbf{m}, \boldsymbol{\Sigma})$ | Gaussian distribution with mean $\mathbf{m} \in \mathbb{R}^d$ and covariance $\boldsymbol{\Sigma} \in \mathbb{R}^{d \times d}$ |
| | $P \otimes Q$ | The product distribution |
| | $I(W; S)$ | The mutual information between $W$ and $S$ |
| | $\mathbb{E}_x$ | The expectation w.r.t. $x$ |
| | $\mathbb{E}$ | The expectation w.r.t. all randomness |
| | $\mathrm{Var}[\nabla f(W, B)\|W]$ | The gradient variance w.r.t. $B$ conditioned by $W$ |
| | $\mathrm{Var}[\nabla f(W, B)]$ | The gradient variance w.r.t. $B$ |
| | $\mathrm{KL}(P\|Q)$ | The Kullback–Leibler divergence of $P$ from $Q$ |
| | $\rho_t$ | The density of $P_{W_t\|S}$ |
| | $\gamma_t$ | The density of $P_{W_t\|S'}$ |
| | $\pi$ | The Gibbs distribution (stationary distribution of the continuous Langevin diffusion) |
| Functions | $l : \mathcal{W} \times \mathcal{Z} \to \mathbb{R}$ | an original loss function |
| | $f : \mathcal{W} \times \mathcal{Z} \to \mathbb{R}$ | a surrogate loss function |
| | $L_\mu, F_\mu$ | The population risk based on an original or a surrogate loss |
| | $L_S, F_S$ | The empirical risk based on an original or a surrogate loss |
| | $F(w, B)$ | The empirical risk with $l$ or $f$ on a mini-batch $B$ |
| | $\nabla F(w, B)$ | The gradient of $F(w, B)$ w.r.t. $w$ |
| | $\mathrm{gen}(\mu, P_{W\|S}; L), \mathrm{gen}(\mu, P_{W\|S}; F)$ | The expected generalization error based on an original or a surrogate loss |
| | $\mathrm{Excess}(\mu, P_{W\|S})$ | The excess risk defined as $\mathbb{E}_{W,S}[F_\mu(W) - F_\mu(w^*)]$ |

# B  Additional information for dissipative losses

Here, we provide the additional information for losses with dissipativity in Assumption 3.

The dissipative assumption plays an essential role in guaranteeing the geometrical convergence of SGLD to the stationary distribution. We note that convergence to the stationary distribution is crucial for reducing training error since the stationary distribution in this context corresponds to the Gibbs posterior distribution of the given loss function. The dissipative assumption is widely used in the research on sampling or non-convex potential function optimization; thus, it is a fundamental property that enables optimization with SGLD rather than strong constraint conditions for generalization. As Mou et al. [24] discussed, the dissipative assumption is weaker than convexity and strong convexity.

Many non-convex losses commonly used in practice satisfy the dissipative property. First, all strongly convex and convex losses obviously satisfy dissipativity. The dissipative losses also include losses that are strongly convex or convex when sufficiently far from zero, that is, there exists $m, R > 0$ such that $\|x - y\|^2 \geq R$, $(x - y) \cdot (\nabla f(x, z) - \nabla f(y, z)) \geq m\|x - y\|^2$ or that $\|x - y\|^2 \geq R$, $(x - y) \cdot (\nabla f(x, z) - \nabla f(y, z)) \geq 0$ for all $x$ and $y$ (refer to Mou et al. [24]). This means that the dissipative losses include the non-convex losses that have a local optimum somewhat close to zero and losses whose tail behavior is similar to the strongly convex losses. A typical example of losses that satisfy dissipativity is non-convex losses with $l_2$ regularization that used for many machine learning models including deep learning models (see Mou et al. [23]). Because of its capability to handle many non-convex losses, the dissipative condition is often employed in the theoretical analysis of non-convex optimization, such as Raginsky et al. [29] and Xu et al. [41].

In the Bayesian context, for example, we often use the negative log-likelihood losses, which satisfy the dissipative property if the likelihood distribution satisfies the Poincaré inequality [2, 32]. Poincaré inequality is applicable to a wide range of practical likelihood distributions, such as log-concave distributions, distributions obtained via bounded perturbations of Poincaré-inequality-satisfying (PIS) distributions, distributions with Gaussian convolution added to bounded losses, distributions formed by Lipschitz continuous transformations of PIS distributions, and direct sums of PIS distributions [2]. Therefore, the dissipative assumption covers many useful Bayesian models, including Bayesian deep learning models [32, 24].

In essence, the dissipative assumption allows for the broad treatment of not only general non-convex optimization in (deep) machine learning but also non-convex losses used in Bayesian inference and Bayesian machine learning. On the other hand, it is essential to note that thick-tailed losses, such as long-tailed $t$-distributions or Cauchy distributions, cannot be handled as a dissipative loss [24].

## C   Difference from generalization error bounds based on uniform convergence

In this section, we discuss the difference between the generalization error bounds based on the information-theoretic (IT) approach and that on the basis of the uniform convergence (UC) notion.

The generalization error bound based on the UC notion guarantees that the generalization error of all hypotheses in the algorithm's output space simultaneously vanishes as the size of the training data increases, ensuring the convergence of generalization error. Furthermore, this bound asserts that within the empirical risk minimization (ERM) principle, it suffices to output any hypothesis from the class that minimizes empirical risk, and by measuring model complexities such as VC-dimension or Rademacher complexity, one can evaluate generalization performance. In other words, the UC-based generalization bounds offer non-trivial guarantees only when the hypothesis class utilized by the algorithm, along with its complexity, is moderately constrained.

On the other hand, deep neural networks (DNNs) are included in vast hypothesis classes where the model complexity drastically increases with the size and depth of the network. When applied to such models, UC-based bounds turn into a vacuous metric due to the exceedingly large model complexity. Furthermore, bounds based on UC rely solely on the hypothesis space and are unable to leverage beneficial statistics obtained from algorithms or datasets, which sometimes results in an inability to capture the true essence of generalization performance. For instance, the gradient variance w.r.t. model parameters exhibit a strong correlation with the generalization performance of deep learning [18]; however, this correlation cannot be represented within UC-type bounds (see Amit et al. [1] for details). This observation leads to the recent interest in data and algorithmic-dependent generalization bounds, such as the PAC-Bayes and IT-based generalization error bounds.

The strength of the IT-based analysis lies in its capacity to directly incorporate the algorithm- and data-dependent statistics related to the generalization performance, such as the gradient variance instead of the model complexity, into the generalization error upper bounds. Especially, the gradient variance is empirically known to have a stronger correlation with the generalization performance of DNNs [18] in comparison to statistics appearing in uniform convergence analysis contexts (e.g., VC dimension, the number of parameters $d$, and the norm of parameters). Although the gradient variance implicitly depends on $d$, it is widely recognized that, in practice, the gradient variance becomes reasonably small as the training proceeds [18].

In our bounds, such as Theorem 4, the generalization error bound is expressed through a quantity that reflects the stability w.r.t. variations in the training data, which is closely related to the generalization properties [23, 25, 35]. While this quantity is expressed via the expectation of gradients and thus is implicitly dependent on $d$ like the gradient variance, it is expected to decrease as training progresses and generalization performance is being enhanced [23, 25] [7]. Consequently, IT-based bounds offer a sensible generalization bound even in models with significantly high complexity, such as DNNs. This is why it has gained attention in the context of SGLD's generalization analysis.

In short, the core aim of IT-based analysis is to offer practical bounds that effectively account for models with high complexity, like DNNs, by directly integrating empirically validated statistics associated with generalization obtained from datasets and an algorithm. Active discussions within

---

[7]   Note that some constants in our bounds explicitly depends on $d$ under non-convex and disspative losses. Removing this dependency is our significant future work. As can be seen in Appendix F.3, however, this dependence does not occur in the convex loss case.

the realm of IT-based analysis revolve around how to analyze the generalization performance of DNNs, which involve non-convex losses, to derive bounds that lead to an accurate understanding of generalization performance.

## D Further discussion for convergence and dependence on dimensions

Here, we provide further discussions on IT-based bounds including ours from the perspectives of convergence and dependence on parameter dimension especially focusing on convex losses.

### D.1 On the convergence of SGLD and our bounds in convex losses

As shown by Shalev-Shwartz et al. [31], there is a convex problem in which the unique solution of ERM fails to generalize. This shows that any optimization algorithm executed for an infinite algorithm iteration must fall into one of two categories: either it never converges to the minimum, or it fails to generalize. The SGLD algorithm leads to the former phenomenon because the obtained parameters via SGLD do not converge to the (local) minima under a fixed temperature parameter $\beta$ for the Gaussian noise coefficient even after infinite iterations.

SGLD rather ensures convergence to a stationary distribution, known as the Gibbs posterior distribution $\pi(\mathrm{d}w) \propto e^{-\beta F(W,S)}$, when $\beta$ remains fixed over time steps. In essence, the trajectory of parameters via the SGLD algorithm gets closer to the minima and then explores its vicinity due to the addition of Gaussian noise to the gradient. Therefore, while convergence to a target distribution occurs, convergence to the minima itself is not achieved without controlling the noise via $\beta$.

Although SGLD does not converge to the minima, it boasts a distinct edge in its ability to explore parameters globally, even within non-convex problems, thanks to the Gaussian noise. This property enables the evaluation of how the obtained expected loss w.r.t. the stationary distribution deviates from that with the global minima. Specifically, we can evaluate this difference by factors that depend on parameter dimensions $d$, $\beta$, and the constants appearing in the assumptions for the potential function, such as dissipativity and smoothness, as elaborated in Appendix G.4. Furthermore, we can also derive the upper bounds for the population risk and excess risk both for convex and non-convex losses (see Raginsky et al. [29], Xu et al. [41]).

### D.2 On dependence on parameter dimension of IT-based bounds

Recently, Livni [21] has shown that every algorithm that guarantees non-trivial population loss on convex problems, must carry dimension-dependent information on the sample. Together with our Theorem 4, this fact implies that, if the temperature $\beta$ is dimension independent, then SGLD will not achieve non-trivial population loss on the (convex) construction in Livni [21]. Alternatively, one could choose dimension-dependent $\beta$ in SGLD but then algorithmic-independent generalization bounds can be easily (and have been) obtained via standard uniform convergence argument.

Unfortunately, removing the dependence on the parameter dimension $d$ is difficult or unavoidable even if our framework is utilized when analyzing the generalization error of discretized Langevin dynamics such as SGLD through the MI between the dataset and parameters. On the other hand, existing and our IT-based bounds such as Theorems 2 and 5 are expressed by the gradient variance or the stability of the expected gradient, which implicitly depend on $d$ but could be smaller than it as training proceeds and the generalization performance is enhanced. We refer to Appendix C for an explanation of the advantages of this property in the IT-based bounds.

In order to theoretically mitigate this reliance on dimensionality, it could be imperative to explore an alternative approach to evaluating generalization that deviates from the MI between parameters and data, which forms the cornerstone of this paper. One possible avenue is, for instance, the utilization of conditional mutual information (CMI) involving super-samples (e.g., Wang and Mao [38]), as highlighted in Livni [21], as well as methods to quantify the MI between the learned hypothesis and dataset (hMI), instead of focusing on the parameters [15]. However, the drawback of these approaches is that it becomes challenging to explicitly incorporate statistics directly obtained from algorithms, such as the gradient variance, into the understanding and evaluation of generalization despite being analyses of algorithm-dependent generalization performance. Seeking IT-based bounds that not only represent algorithm- and data-dependent statistics related to generalization performance, such as

gradient variance but also theoretically eliminate dimension dependence constitutes a significant future work in the context of the IT-based generalization analysis field.

# E    Theoretical properties of SGLD and Langevin diffusion

Here, we show some theoretical properties of SGLD under Assumptions 2 and 3.

**Lemma 2** (Adapted from Farghly and Rebeschini [10]). *Suppose that Assumptions 2 and 3 are satisfied. Then, for any $z \in \mathcal{Z}$, we have*

$$\|\nabla f(0, z)\| \leq M\sqrt{\frac{b}{m}}.$$

*Proof.* We straightforwardly obtain the above claim from Assumptions 2 and 3 with $w = 0$. $\qquad\square$

**Lemma 3** (Modified version from Raginsky et al. [29]). *Suppose that Assumption 2 is satisfied. Then, for any $z \in \mathcal{Z}$ and all $w \in \mathcal{W}$, we have*

$$\|\nabla f(w, z)\| \leq M\|w\| + M\sqrt{\frac{b}{m}}.$$

*Proof.* Raginsky et al. [29] derived the upper bound of the gradient as $\|\nabla f(w, z)\| \leq M\|w\| + B$ by assuming the the following condition: $\|\nabla f(0, z)\| \leq B$ $(B > 0)$. We replace the constant $B$ by $M\sqrt{\frac{b}{m}}$ based on Lemma 2. $\qquad\square$

**Lemma 4** (Modified version from Xu et al. [41]). *Suppose that Assumptions 2 and 3 are satisfied. Then, for any $z \in \mathcal{Z}$, we have*

$$\mathbb{E}\|\nabla F_S(w) - \nabla F(w, B)\|^2 \leq \frac{8(n-k)M^2(\|w\|^2 + \frac{k}{m})}{k(n-1)} := 8\delta M^2\left(\|w\|^2 + \frac{k}{m}\right),$$

*where $\delta := \frac{n-k}{k(n-1)} \in (0, 1]$.*

*Proof.* Xu et al. [41] assumed that $\nabla F_S(w)$ is dissipative. In contrast, we posed the dissipative assumption on $\nabla f(w, z)$ for each $z$ following Farghly and Rebeschini [10]. We then modified the upper bound of the stochastic gradient shown in Xu et al. [41]. $\qquad\square$

**Lemma 5** (Modified version from Raginsky et al. [29] and Xu et al. [41]). *Suppose that Assumptions 2 and 3 are satisfied. Let $\eta \in (0, 1 \wedge \frac{m}{5M^2})$ be fixed. Then, for any $z \in \mathcal{Z}$ and any $t \in \mathbb{N}$, we have*

$$\mathbb{E}\|W_{t+1}\|^2 \leq (1 - 2\eta m + 10\eta^2 M^2)\mathbb{E}\|W_t\|^2 + 2\eta\left(b + 10\eta M^2\frac{b}{m} + \frac{d}{\beta}\right),$$

*and*

$$\mathbb{E}\|W_{t+1}\|^2 \leq \begin{cases} 2\eta(b + 10\eta M^2\frac{b}{m} + \frac{d}{\beta}) & (A_{\eta,m,M} \leq 0) \\ (1 - 2\eta m + 10\eta^2 M^2)^t\mathbb{E}\|W_0\|^2 + 2\frac{b+10\eta M^2\frac{b}{m}+\frac{d}{\beta}}{m-5\eta M^2} & (0 \leq A_{\eta,m,M} \leq 1), \end{cases} \tag{19}$$

*where $A_{\eta,m,M} := (1 - 2\eta m + 10\eta^2 M^2)$. Combining the inequalities in Eq. (19), we have*

$$\mathbb{E}\|W_t\|^2 \leq \mathbb{E}\|W_0\|^2 + 2\left(1 \vee \frac{1}{m}\right)\left(b + 10\eta M^2\frac{b}{m} + \frac{d}{\beta}\right)$$

$$\leq s^2 + 2\left(1 \vee \frac{1}{m}\right)\left(b + 10\eta M^2\frac{b}{m} + \frac{d}{\beta}\right) =: C_0,$$

*where $C_0$ is independent of $\eta$ and $\beta$ and $s^2$ is the square moment of the initial distribution.*

*Proof.* We slightly modified the coefficients of the upper bound of the $l_2$ norm of the parameter shown in Raginsky et al. [29] and Xu et al. [41] based on the upper bound of the stochastic gradient Lemma 4. $\qquad\square$

Combining Lemmas 3 and 5, we have the following upper bound for the stochastic gradient.

**Lemma 6.** *Suppose that Assumptions 2 and 3 are satisfied. Let $\eta \in (0, 1 \wedge \frac{m}{5M^2})$ be fixed. Then, for any $z \in \mathcal{Z}$ any $t \in \mathbb{N}$, we have*

$$\mathbb{E}\|\nabla F(W_t, B)\|^2 \leq$$
$$\begin{cases} 2M^2\eta^2(b + 4\eta M^2\frac{b}{m} + \frac{d}{\beta}) + M^2\frac{b}{m} & (A_{\eta,m,M} \leq 0) \\ M^2(1 - 2\eta m + 10\eta^2 M^2)^t\mathbb{E}\|W_0\|^2 + 2M^2\frac{b+4\eta M^2\frac{b}{m}+\frac{d}{\beta}}{m-\eta M^2} + M^2\frac{b}{m} & (0 \leq A_{\eta,m,M} \leq 1), \end{cases}$$

*where $A_{\eta,m,M} := (1 - 2\eta m + 10\eta^2 M^2)$. Combining the above inequalities, we have*

$$\mathbb{E}\|\nabla F(W_t, B)\|^2 \leq M^2 C_0 + M^2\frac{b}{m}.$$

*Proof.* We obtain the result by combining Lemma 3 and 5 and using the Jensen inequality. $\qquad\square$

## F  Proofs of the generalization error bound with surrogate loss in Section 3

This section provides the complete proof of Theorem 4 restated as follows.

**Theorem 4.** *Suppose that Assumptions 1, 2, 3, and 4 are satisfied. Then, for any $\beta \geq 2/m$ and $\eta \in (0, 1 \wedge \frac{m}{5M^2} \wedge 4\beta c_{\mathrm{LS}})$ and any $T \in \mathbb{N}$, we have*

$$|\mathrm{gen}(\mu, P_{W_T|S}; L)| \leq \sqrt{\frac{2c_1\sigma_g^2}{n}\left(1 \wedge \frac{\eta T}{4\beta c_{\mathrm{LS}}}\right)(V_\nabla + c_2)}, \tag{6}$$

*where $c_1$, $c_2$, and $V_\nabla$ are the positive constant terms w.r.t. $\{M, m, b, d, \beta, s^2\}$.*

Our proof consists of the following three steps. We construct the FP equations for the density of the parameters under two different datasets and derive the time evolution of the KL divergence as the upper bound of the MI (Appendix F.1). We then analyze this time evolution by using the *parametrix method* (Appendix F.2) for solving the FP equation. The distinction between this section and Section 3.3 lies in the focus of SGLD, which employs stochastic gradients and random noise from $\mathcal{N}(0, \mathbf{I}_d)$, as opposed to the continuous Langevin diffusion that employs full-batch gradients and standard Brownian motion.

### F.1  FP equation for SGLD and time evolution of the KL divergence

As the first step, we construct the two different FP equations for the parameter density.

We define a one-step SGLD at the initial step as follows:

$$\mathrm{d}W_t = -\nabla F(W_0, B_0)\mathrm{d}t + \sqrt{2\beta^{-1}}\mathrm{d}H_t.$$

Note that, at time $t = \eta$,

$$W_\eta = W_0 - \eta\nabla F(W_0, B_0) + \sqrt{2\eta\beta^{-1}}H_\eta,$$

is distributionally equivalent to

$$W_\eta = W_0 - \eta\nabla F(W_0, B_0) + \sqrt{2\eta\beta^{-1}}\xi,$$

where $\xi \sim N(0, \mathbf{I}_d)$.

The distribution $\rho_t$ of $W_t$ depends on random variables $W_0$ and $B_0$. We thus denote the joint distribution of $\{W_0, W_t, B_0\}$ under a dataset $S$ as $\rho_{0tB}(W_0, W_t, B_0)$, where $\rho_0$ is the distribution of $W_0$ and $U$ is the *uniform* distribution of $B_0$. Then, its conditional and marginal distribution is expressed as

$$\rho_{0tB}(W_0, W_t, B_0) = \rho_0(w_0)U(B_0)\rho_{t|0,B}(W_t|W_0, B_0) = \rho_{tB}(W_t, B)\rho_{0|t,B}(W_0|W_t, B_0).$$

Since we have introduced so many notations, for the sake of simplicity, we allow the abuse of notation and let $\rho_{t|0B}$ and $\rho_t$ denote both the distribution and density.

On the basis of these facts, we can obtain the FP equation for $\rho_{t|0B}(w_t)$ as

$$\frac{\partial \rho_{t|0B}}{\partial t} = \nabla \cdot \left( \frac{1}{\beta} \nabla \rho_{t|0B} + \rho_{t|0B} \nabla F(w_0, B_0) \right), \tag{20}$$

and its marginal process as

$$\frac{\partial \rho_t}{\partial t} = \nabla \cdot \left( \frac{1}{\beta} \nabla \rho_t + \rho_t \mathbb{E}_{\rho_{0B|t}}[\nabla F(w_0, B_0)|w_t = w] \right), \tag{21}$$

which is derived in Vempala and Wibisono [32] and Kinoshita and Suzuki [19]. As shown above, the randomness associated with the dataset can be handled by simply taking the expectation for conditional gradients with respect to a uniform distribution $U$. To avoid cumbersome discussions, we omit descriptions related to the expectation with respect to $B_0$ from here on.

As with the first step, we can define a one-step SGLD with the joint density $\gamma_{0t}(w_0, w_t)$ and the conditional distribution $\gamma_0(w_0)\gamma_{t|0}(w_t|w_0)$ under a dataset $S'(\neq S)$, where $\gamma_0(w_0)\gamma_{t|0}(w_t|w_0)$ corresponds to the marginal distribution, i.e., $\gamma_0(w_0)\gamma_{t|0}(w_t|w_0) = \gamma_t(w_t)\gamma_{0|t}(w_0|w_t)$. We also can obtain the FP equation and its marginal process in the form of $\rho_{t|0}$ and $\rho_t$ replaced by $\gamma_{t|0}$ and $\gamma_t$ in Eqs. (20) and (21).

In Section 3.3, we obtain the upper bound of the MI as follows:

$$I(W_t; S) \leq \mathbb{E}_{S,S'}\mathrm{KL}(P_{W_t|S}|P_{W_t|S'}) = \mathbb{E}_{S,S'}\mathrm{KL}(\rho_t|\gamma_t).$$

By taking the derivation w.r.t. $t$, we have

$$\frac{\partial \mathrm{KL}(\rho_t|\gamma_t)}{\partial t} = \int \mathrm{d}w \left( \frac{\partial \rho_t}{\partial t} \log \frac{\rho_t}{\gamma_t} \right) - \int \mathrm{d}w \left( \frac{\rho_t}{\gamma_t} \frac{\partial \gamma_t}{\partial t} \right). \tag{22}$$

The first and second terms can be expressed as

$$\int \mathrm{d}w \left( \frac{\partial \rho_t}{\partial t} \log \frac{\rho_t}{\gamma_t} \right) = -\frac{1}{\beta} \int \mathrm{d}w \nabla \log \rho_t \cdot \nabla \log \frac{\rho_t}{\gamma_t} - \int \mathrm{d}w \rho_{0t} \nabla \log \frac{\rho_t}{\gamma_t} \cdot \nabla F(w_0, B_0),$$

and

$$\int \mathrm{d}w \left( \frac{\rho_t}{\gamma_t} \frac{\partial \gamma_t}{\partial t} \right) = -\frac{1}{\beta} \int \mathrm{d}w \nabla \frac{\rho_t}{\gamma_t} \cdot \nabla \gamma_t - \int \mathrm{d}w \frac{\rho_t}{\gamma_t} \nabla \log \frac{\rho_t}{\gamma_t} \cdot \gamma_{0t} \nabla F(W_0', B_0'),$$

where $W_0'$ follows $\gamma_0$, which is the density of the initial distribution and $B_0'$ is the stochastic gradient based on $S'$.

According to these facts, Eq. (22) can be rewritten as

$$\frac{\partial \mathrm{KL}(\rho_t|\gamma_t)}{\partial t}$$

$$= -\frac{1}{\beta} \mathbb{E}_{\rho_t} \|\nabla \log \rho_t - \nabla \log \gamma_t\|^2$$

$$\qquad - \int \rho_{0t} \mathrm{d}w \nabla \log \frac{\rho_t}{\gamma_t} \cdot \nabla F(W_0, B_0) + \int \mathrm{d}w \frac{\rho_t}{\gamma_t} \nabla \log \frac{\rho_t}{\gamma_t} \cdot \gamma_{0t} \nabla F(W_0', B_0')$$

$$= -\frac{1}{\beta} \mathbb{E}_{\rho_t} \|\nabla \log \rho_t - \nabla \log \gamma_t\|^2 - \int \rho_t \mathrm{d}w \nabla \log \frac{\rho_t}{\gamma_t} \cdot \mathbb{E}_{\rho_{0|t}}[\nabla F(w_0, B_0)|W_t = w]$$

$$\qquad + \int \mathrm{d}w \frac{\rho_t}{\gamma_t} \nabla \log \frac{\rho_t}{\gamma_t} \cdot \gamma_t \mathbb{E}_{\gamma_{0|t}}[\nabla F(w_0', B_t')|W_t = w)]$$

$$= -\frac{1}{\beta} \mathbb{E}_{\rho_t} \|\nabla \log \rho_t - \nabla \log \gamma_t\|^2$$

$$\qquad - \int \rho_t \mathrm{d}w \nabla \log \frac{\rho_t}{\gamma_t} \cdot (\mathbb{E}_{\rho_{0|t}}[\nabla F(W_0, B_0)|W_t = w] - \mathbb{E}_{\gamma_{0|t}}[\nabla F(W_0', B_t')|W_t = w])$$

$$\leq -\frac{1}{2\beta} \mathbb{E}_{\rho_t} \|\nabla \log \rho_t - \nabla \log \gamma_t\|^2$$

$$\qquad + \frac{\beta}{2} \int \rho_t \mathrm{d}w \|\mathbb{E}_{\rho_{0|t}}[\nabla F(W_0, B_0)|W_t = w] - \mathbb{E}_{\gamma_{0|t}}[\nabla F(W_0', B_t')|W_t = w]\|^2,$$

where the final inequality comes from the Cauchy–Schwartz inequality. We define $\widetilde{V}_{\nabla_t} := \int \rho_t dw \|\mathbb{E}_{\rho_{0|t}}[\nabla F(W_0, B_0)|W_t = w] - \mathbb{E}_{\gamma_{0|t}}[\nabla F(W_0', B_t')|W_t = w]\|^2$ for simplicity. We evaluate this term in Appendix F.2.3.

In the same way as Section 3.3, we have the following inequality by introducing the logarithm of the stationary distribution $\nabla \log \pi(w)$ into $\mathbb{E}\|\nabla \log \rho_t - \nabla \log \gamma_t\|^2$ in the above:

$$
\begin{aligned}
&- \mathbb{E}_{\rho_t}\|\nabla \log \rho_t - \nabla \log \gamma_t\|^2 \\
&= -\mathbb{E}_{\rho_t}\|\nabla \log \rho_t\|^2 - \mathbb{E}_{\rho_t}\|\nabla \log \gamma_t\|^2 + 2\mathbb{E}_{\rho_t}\nabla \log \rho_t \cdot \nabla \log \gamma_t \\
&\leq -\frac{1}{2}\mathbb{E}_{\rho_t}\|\nabla \log \rho_t - \nabla \log \pi\|^2 + \mathbb{E}_{\rho_t}\|\nabla \log \pi\|^2 + 2\mathbb{E}_{\rho_t}\nabla \log \rho_t \cdot \nabla \log \gamma_t,
\end{aligned}
$$

where we used $-x^2 \leq -\|x - y\|^2/2 + y^2$ for all $x, y \in \mathbb{R}^d$. To simplify the notation, we express the second and third terms as $\Omega(\rho_t, \gamma_t, \pi) := \mathbb{E}_{\rho_t}\|\nabla \log \pi\|^2 + 2\mathbb{E}_{\rho_t}\nabla \log \rho_t \cdot \nabla \log \gamma_t$. From the above fact, we obtain

$$
\begin{aligned}
\frac{\partial \mathrm{KL}(\rho_t|\gamma_t)}{\partial t} &\leq -\frac{1}{4\beta}\mathbb{E}_{\rho_t}\|\nabla \log \rho_t - \nabla \log \pi\|^2 + \frac{1}{2\beta}\Omega(\rho_t, \gamma_t, \pi) + \frac{\beta}{2}\widetilde{V}_{\nabla_t} \\
&\leq -\frac{1}{4\beta c_{\mathrm{LS}}}\mathrm{KL}(\rho_t|\pi) + \frac{1}{2\beta}\Omega(\rho_t, \gamma_t, \pi) + \frac{\beta}{2}\widetilde{V}_{\nabla_t} \\
&\leq -\frac{1}{4\beta c_{\mathrm{LS}}}\left(\mathrm{KL}(\rho_t|\gamma_t) + \mathbb{E}_{\rho_t}\log\frac{\gamma_t}{\pi}\right) + \frac{1}{2\beta}\Omega(\rho_t, \gamma_t, \pi) + \frac{\beta}{2}\widetilde{V}_{\nabla_t}, \qquad (23)
\end{aligned}
$$

where the second inequality is from the LSI [2]. In the above, the LSI constant $c_{\mathrm{LS}}$ is defined by Bakry et al. [2] as follows:

$$
\begin{aligned}
c_{\mathrm{LS}} &\leq \lambda_l := 2D_1 + 2\rho_0^{-1}(D_2 + 2), \\
\rho_0^{-1} &\leq \frac{2C(d + b\beta)}{m\beta}\exp\left(\frac{2}{m}(M + B)(b\beta + d) + \beta(A + B)\right) + \frac{1}{m\beta(d + b\beta)},
\end{aligned}
$$

where $D_1 = \frac{2m^2 + 8M^2}{\beta m^2 M}$, $D_2 \leq \frac{6M(d+\beta)}{m}$, and $C$ is the universal constant (see also Appendices B and E in Raginsky et al. [29]).

Multiplying $e^{\frac{t}{4\beta c_{\mathrm{LS}}}}$ for both hands in Eq. (23) yields

$$
e^{\frac{t}{4\beta c_{\mathrm{LS}}}}\frac{\partial \mathrm{KL}(\rho_t|\gamma_t)}{\partial t} \leq -e^{\frac{t}{4\beta c_{\mathrm{LS}}}}\frac{1}{4\beta c_{\mathrm{LS}}}\mathbb{E}_{\rho_t}\log\frac{\gamma_t}{\pi} + \frac{1}{2\beta}e^{\frac{t}{4\beta c_{\mathrm{LS}}}}\Omega(\rho_t, \gamma_t, \pi) + \frac{\beta}{2}\widetilde{V}_{\nabla_t}.
$$

By integrating time $t = 0 \to \eta$, we obtain

$$
\begin{aligned}
e^{\frac{\eta}{4\beta c_{\mathrm{LS}}}}\mathrm{KL}(\rho_\eta|\gamma_\eta) &\leq \mathrm{KL}(\rho_0|\gamma_0) - \int_0^\eta dt\, e^{\frac{t}{4\beta c_{\mathrm{LS}}}}\frac{1}{4\beta c_{\mathrm{LS}}}\mathbb{E}_{\rho_t}\log\frac{\gamma_t}{\pi} \\
&\quad + \int_0^\eta dt\frac{1}{2\beta}e^{\frac{t}{4\beta c_{\mathrm{LS}}}}\Omega(\rho_t, \gamma_t, \pi) + \frac{\beta}{2}\int_0^\eta dt\, e^{\frac{t}{4\beta c_{\mathrm{LS}}}}\widetilde{V}_{\nabla_t}. \qquad (24)
\end{aligned}
$$

We later evaluate the second and third terms of Eq. (24) in Appendix F.2. We then evaluate the fourth term $\widetilde{V}_{\nabla_t}$ of Eq. (24) as $\widetilde{V}_{\nabla_t} \leq D_1$, where $D_1$ is problem dependent constant that can be independent of $\eta$, in Appendix F.2.3.

## F.2   The solution for the FP equation via the parametrix method

In this section, we evaluate the second and third terms in Eq. (24) by utilizing the *parametrix method* for the FP equation [12, 27].

### F.2.1   Consequences of the parametrix method for the FP equation

We summarize two essential consequences of the parametrix method used in Appendix F.2.2.

**Solution expansion of the FP equation.** The first consequence of this method is as follows. Given the following FP equation,

$$\nabla \cdot \left( \frac{1}{\beta} \nabla \rho_t(w) + \rho_t(w) b(w,t) \right) - \frac{\partial \rho_t(w)}{\partial t} = 0,$$

and initial condition $\rho_{t=0} = \rho_0$, the solution can be expanded as

$$\rho_t(w) = \mathbb{E}_{\xi \sim \rho_0} Z(w,t;\xi,0) + \mathbb{E}_{\xi \sim \rho_0} \int_0^t \int_{\mathbb{R}^d} \mathrm{d}z Z(w,t;z,\tau) \Phi(z,\tau;\xi,0), \qquad (25)$$

where

$$Z(x,t;z,\tau) := \frac{1}{(\frac{4\pi}{\beta}(t-\tau))^{d/2}} e^{-\frac{\beta\|x-z\|^2}{4(t-\tau)}},$$

and

$$\Phi(z,\tau;\xi,t) := \sum_{n=1}^{\infty} L^n Z(z,\tau;\xi,t). \qquad (26)$$

In Eq. (26), $L^n Z(z,\tau;\xi,t)$ is defined through

$$L^{n+1} Z(x,\tau;\xi,t) := \int_t^{\tau} \int_{\mathbb{R}^d} \mathrm{d}s\mathrm{d}y (LZ(x,\tau;y,s))(L^n Z(y,s;\xi,t)),$$

where

$$LZ(x,\tau;\xi,t) := b(x,\tau) \cdot \nabla_x Z(x,\tau;\xi,t),$$

and thus $L^1 Z(x,\tau;\xi,t) = LZ(x,\tau;\xi,t)$.

The above expansion requires the convergence of Eq. (26). Fortunately, this condition holds for the Langevin diffusion, for example, because $\rho_0(w)$ and $b(w,t) = \nabla F(w,S)$ satisfies the following two assumptions for the initial state and $b(w,t)$ from Lemma 3 and Assumption 4: (i) there exist positive constants $a$ and $b$ such that $\rho_0(w) \leq ae^{b\|w\|^2} < \infty$ for all $w \in \mathcal{W}$, and (ii) there exist some positive constants $a'$ and $b'$ such that $\|b(w,t)\| < a'\|w\| + b'$ for all $w \in \mathcal{W}$.

**Parametrix solution is twice differentiable.** Another important consequence is that the parametrix solution $\rho_t(w)$ is twice differentiable with respect to $w$. Under the initial distribution $\mathcal{N}(0, s^2 \mathbf{I}_d)$ with Assumption 4, we can obtain the following facts according to Friedman [12] and Pavliotis [27]:

$$\rho_t(w) \leq \frac{1}{(2\pi(s^2 + \frac{2t}{\beta}))^{d/2}} e^{-\frac{\|w\|^2}{2(s^2 + \frac{2t}{\beta})}} + \left( s^2 + \frac{2t}{\beta} \right)^{1/2} \frac{C_0}{(2\pi(s^2 + \frac{2t}{\beta}))^{d/2}} e^{-\frac{\|w\|^2}{2(s^2 + \frac{2t}{\beta})}}, \quad (27)$$

$$\sum_{i=1}^d \left| \frac{\partial \rho_t(w)}{\partial w_i} \right| \leq \frac{C_1}{(2\pi(s^2 + \frac{2t}{\beta}))^{(d+1)/2}} e^{-\frac{\|w\|^2}{2(s^2 + \frac{2t}{\beta})}} + \frac{C_2}{(2\pi(s^2 + \frac{2t}{\beta}))^{d/2}} e^{-\frac{\|w\|^2}{2(s^2 + \frac{2t}{\beta})}}, \qquad (28)$$

and

$$\sum_{i,j=1}^d \left| \frac{\partial^2 \rho_t(w)}{\partial w_i \partial w_i} \right| \leq \frac{C_3}{(2\pi(s^2 + \frac{2t}{\beta}))^{(d+2)/2}} e^{-\frac{\|w\|^2}{2(s^2 + \frac{2t}{\beta})}}$$

$$+ \left( s^2 + \frac{2t}{\beta} \right)^{-1/2} \frac{C_4}{(2\pi(s^2 + \frac{2t}{\beta}))^{d/2}} e^{-\frac{\|w\|^2}{2(s^2 + \frac{2t}{\beta})}}, \quad (29)$$

where $\{C_0, C_1, C_2, C_3, C_4\}$ are positive constants w.r.t. $\{m, M, \beta, d, b, s^2\}$.

Eqs. (27) and (28) can be derived by following the proof of Theorem 11 in Friedman [12]. The statement of this theorem is about the transition kernel; therefore, it corresponds to the case where the expectation with respect to $\xi \sim \rho_0$ is excluded from Eq. (25). In light of this fact, we take the convolution by the initial distribution $\xi \sim \rho_0$ for the beginning part of the proof of Theorem 11.

This approach is equivalent to taking convolutions in the overall discussion of Section 4 described by Friedman [12]. After convolution, following the proof of Theorem 11 leads to Eqs. (27) and (28). Theorem 11 does not yield results related to second-order differentials; however, we can derive Eq. (29) by combining Lemma 3 from Friedman [12] into the proof of Theorem 11 and employing a similar way as described above. While it is assumed that the coefficients of the FP equation are bounded in Friedman [12], we can relax this assumption to unbounded coefficients as shown in subsequent work such as Deck and Kruse [7].

### F.2.2 Applying the parametrix method for SGLD's FP equations

Now, we get back to the SGLD setting. First of all, it should be mentioned that we do not lose generality by focusing solely on the initial iteration, i.e., $t = 0 \to \eta$. This reason is as follows.

For the initial iteration ($t = 0 \to \eta$), we can see that $b(x, t) = \mathbb{E}_{\rho_{0|t}}[\nabla F(W_0, B_0)|W_t = w]$ from Eq. (21). As we explained in Appendix F.2.1, the condition of the expansion is satisfied under Lemma 3 and Assumption 4. Thus, the solution and its differentiation can be obtained via the parametrix solution, expressed as Eqs. (27), (28), and (29) with the constants $\{C_0, C_1, C_2, C_3, C_4\}$ that depend on the problem except $\eta$. When considering the second iteration ($t = \eta \to 2\eta$), the initial distribution is expressed as $\rho_\eta$. The concern here is whether the solution of the FP for SGLD satisfies the conditions of the parametrix method in this case. Fortunately, these conditions are also satisfied in the second iteration. The initial condition of the expansion is satisfied from Eq. (27), and the condition $b(x, t) = \mathbb{E}_{\rho_{\eta|t}}[\nabla F(W_\eta, B_1)|W_t = w]$ also satisfies the condition of the FP expansion from Lemma 3. We thus have the same form of the solution in Eqs. (27), (28), and (29) at time $t = \eta \to 2\eta$. In the same way, the solution at $t \in (s\eta, (s+1)\eta]$ for $s \in \mathbb{N}$ can be expanded as the same parametrix expansion.

**Bounding $\Omega(\rho_t, \gamma_t, \pi)$ (related to the third term in Eq. (24)).** We can decompose $\Omega(\rho_t, \gamma_t, \pi)$ as

$$\mathbb{E}_{\rho_t}\|\nabla \log \pi\|^2 + 2\mathbb{E}_{\rho_t}\nabla \log \rho_t \cdot \nabla \log \gamma_t = \mathbb{E}_{\rho_t}\|\nabla \log \pi\|^2 + 2\int dw \nabla \rho_t \cdot \nabla \log \gamma_t.$$

To derive the upper bound of the right-hand side, we focus on the following facts:

$$\int dw \nabla \rho_t \cdot \nabla \log \gamma_t = \int dw \sum_{i=1}^d \frac{\partial \rho_t}{\partial w_i} \frac{\partial \log \gamma_t}{\partial w_i}. \tag{30}$$

For the $i$-th dimension, we have

$$\int dw \frac{\partial \rho_t}{\partial w_i} \frac{\partial \log \gamma_t}{\partial w_i} = -\int dw \frac{\partial^2 \rho_t}{\partial w_i^2} \log \gamma_t \leq \int dw \left| \frac{\partial^2 \rho_t}{\partial w_i^2} \right| |\log \gamma_t|, \tag{31}$$

where we used the integration by parts from the fact that $\frac{\partial \rho_t}{\partial w_i} \to 0$ as $\|w\| \to 0$ according to the expansion in Eq. (28). Details of this argument can be found in Mou et al. [24]. By using the result in Eq. (27) for $\rho_t$ and $\gamma_t$, we have

$$\int dw \left| \frac{\partial^2 \rho_t}{\partial w_i^2} \right| |\log \gamma_t| \leq \frac{C_1'}{(2\pi(s^2 + \frac{2t}{\beta}))^{1/2}} + C_2', \tag{32}$$

where the Gaussian integral is used for $\frac{\partial^2 \rho_t}{\partial w_i^2}$ in Eq. (29) and $\rho_t$ in Eq. (27) is replaced to $\gamma_t$. We note that $C_1'$ and $C_2'$ only depend on $\{m, M, \beta, d, b, s^2\}$.

Substituting Eqs. (31) and (32) into Eq. (30), we obtain

$$\int dw \nabla \rho_t \cdot \nabla \log \gamma_t \leq \frac{dC_1'}{(2\pi(s^2 + \frac{2t}{\beta}))^{1/2}} + dC_2'. \tag{33}$$

From Lemma 6, we have

$$\mathbb{E}_{\rho_t}\|\nabla \log \pi\|^2 \leq \beta^2 \mathbb{E}_{\rho_t}\|\nabla F_S(w)\|^2$$

$$\leq \beta^2 M^2 \left( s^2 + 2\left(1 \vee \frac{1}{m}\right)\left(b + 10M^2 \frac{b}{m} + \frac{d}{\beta}\right)\right) + \beta^2 M^2 \frac{b}{m}. \tag{34}$$

Eqs. (33) and (34) leads to $\Omega(\rho_t, \gamma_t, \pi) \leq D_2$, where

$$D_2 := \beta^2 M^2 \left( s^2 + 2\left(1 \vee \frac{1}{m}\right)\left(b + 10M^2 \frac{b}{m} + \frac{d}{\beta}\right) + \frac{b}{m}\right) + \frac{dC_1'}{(2\pi(s^2 + \frac{2t}{\beta}))^{1/2}} + dC_2'. \tag{35}$$

**Bounding** $-\mathbb{E}_{\rho_t}\log\frac{\gamma_t}{\pi}$ **(related to the second term in Eq.** (24)**).** By using the Kolmogorov solution of the FP equation [2], we have

$$\gamma_t(w) = \mathbb{E}_{W_T}[\gamma_0(W_T)|W_0 = w]$$

$$= \mathbb{E}_{W_T}\left[\frac{1}{(2\pi s^2)^{d/2}}e^{-\frac{\|W_T\|^2}{2s^2}}\bigg|W_0 = w\right] \geq \frac{1}{(2\pi s^2)^{d/2}}e^{-\frac{\mathbb{E}_{W_T}[\|W_T\|^2|W_0=w]}{2s^2}}, \quad (36)$$

where the last inequality comes from Jensen's inequality. The above inequality gives us

$$-\mathbb{E}_{\tilde{W}_T}\log\gamma_t(\tilde{W}_T) = -\mathbb{E}_{\tilde{W}_T}\log\mathbb{E}_{W_T}[\gamma_0(W_T)|W_0 = \tilde{W}_T]$$

$$\leq -\mathbb{E}_{\tilde{W}_T}\mathbb{E}_{W_T}[\log\gamma_0(W_T)|W_0 = \tilde{W}_T]$$

$$\leq \frac{d}{2}\log(2\pi s^2) + \frac{1}{2s^2}\mathbb{E}_{\tilde{W}_T}\mathbb{E}_{W_T}[\|W_T\|^2|W_0 = \tilde{W}_T],$$

where $\tilde{W}_T$ is the independent copy of $W_T$ and the first and second inequalities are obtained from Jensen's inequality and Eq. (36), respectively. By using Lemma 5 twice, we obtain

$$-\mathbb{E}_{\tilde{W}_T}\log\gamma_t(\tilde{W}_T) \leq \frac{d}{2}\log(2\pi s^2) + \frac{1}{2s^2}\left(\mathbb{E}_{\tilde{W}_T}\|\tilde{W}_T\|^2 + 2\left(1\vee\frac{1}{m}\right)\left(b + 10M^2\frac{b}{m} + \frac{d}{\beta}\right)\right)$$

$$\leq \underbrace{\frac{d}{2}\log(2\pi s^2) + \frac{1}{2s^2}\left(s^2 + 4\left(1\vee\frac{1}{m}\right)\left(b + 10M^2\frac{b}{m} + \frac{d}{\beta}\right)\right)}_{=:B_1}.$$

In addition, from Lemma 8, we have

$$\mathbb{E}_{\rho_t}\log\pi = \beta M\mathbb{E}_{\rho_t}\|W\|^2 + \frac{\beta b}{2m} + A$$

$$\leq \underbrace{\beta M\left(s^2 + 2\left(1\vee\frac{1}{m}\right)\left(b + 10M^2\frac{b}{m} + \frac{d}{\beta}\right)\right) + \frac{\beta b}{2m} + A}_{=:B_2}.$$

Thus, we have the upper bound of $-\mathbb{E}_{\rho_t}\log\frac{\gamma_t}{\pi}$ as

$$-\mathbb{E}_{\rho_t}\log\frac{\gamma_t}{\pi} \leq B_1 + B_2 =: D_3, \quad (37)$$

where $D_3$ is the positive constant only depends on $\{m, M, \beta, d, b, s^2\}$.

### F.2.3  Bounding the stability term and finalizing proof

Finally, we show the upper bound of the stability term expressed as $\widetilde{V}_{\nabla_t}$ in Eq. (24). Similarly to Appendix F.2.2, we focus on the initial iteration $t = 0 \rightarrow \eta$. From the definition of $\widetilde{V}_{\nabla_t}$, we have

$$\widetilde{V}_{\nabla_t} = \int\rho_t\mathrm{d}w\|\mathbb{E}_{\rho_{0|t}}[\nabla F(W_0, B_0)|W_t = w] - \mathbb{E}_{\gamma_{0|t}}[\nabla F(W_0', B_t')|W_t = w]\|^2$$

$$\leq 2\int\rho_t\mathrm{d}w\|\mathbb{E}_{\rho_{0|t}}[\nabla F(W_0, B_0)|W_t = w]\|^2 + 2\int\rho_t\mathrm{d}w\|\mathbb{E}_{\gamma_{0|t}}[\nabla F(W_0', B_t')|W_t = w]\|^2.$$

The first term of the above can be rewritten as

$$\int\rho_t\mathrm{d}w\|\mathbb{E}_{\rho_{0|t}}[\nabla F(W_0, B_0)|W_t = w]\|^2 \leq \mathbb{E}_{\rho_0}\|\nabla F(W_0, B_0)\|^2,$$

by using Jensen's inequality for the conditional distribution. Since $\mathbb{E}_{\rho_0}\|\nabla F(W_0, B_0)\|^2$ can be bounded by using Eq. (34), we have

$$\int\rho_t\mathrm{d}w\|\mathbb{E}_{\rho_{0|t}}[\nabla F(W_0, B_0)|W_t = w]\|^2$$

$$\leq M^2\left(s^2 + 2\left(1\vee\frac{1}{m}\right)\left(b + 10M^2\frac{b}{m} + \frac{d}{\beta}\right)\right) + M^2\frac{b}{m} =: D_4. \quad (38)$$

Next, we derive the upper bound of

$$\int \rho_t \mathrm{d}w \|\mathbb{E}_{\gamma_{0|t}}[\nabla F(W_0', B_t')|W_t = w]\|^2 = \mathbb{E}_{\rho_t}\mathbb{E}_{\gamma_{0|t}}[\|\nabla F(W_0', B_t')|W_t = w]\|^2.$$

From Lemma 3, we have $\|\nabla F(W_0', B_t')\|^2 \leq 2M^2\|W_0'\|^2 + 2M^2\frac{b}{m}$. Thus, we need to evaluate $\mathbb{E}_{\rho_t}\mathbb{E}_{\gamma_{0|t}}[\|W_0'\|^2|W_t' = w]$; however, it is difficult to analyze this expectation because the densities $\rho$ and $\gamma$ at time $t$ and $0$ are different.

Fortunately, we can circumvent this difficulty by using the reverse process formulae shown in Haussmann and Pardoux [16]. According to the fact that the conditional expectation $\mathbb{E}_{\gamma_{0|t}}[\cdot]$ implies the reverse process of Eq. (20). This formulae gives us the following reverse process for time $s$ $(0 \leq s \leq t)$:

$$\mathrm{d}\tilde{W}_s = [\nabla F(\tilde{W}_{s=t}, B'_{s=t}) + 2\beta^{-1}\nabla \log \gamma_{t-s}]\mathrm{d}t + \sqrt{2\beta^{-1}}\mathrm{d}H_s, \quad \tilde{W}_0 \sim \gamma_t. \tag{39}$$

In the above, $B'_{s=t}$ implies $B'_0$ in the original forward process and thus a mini-batch sample is fixed. We obtain the relationship $\tilde{\gamma}_s = \gamma_{t-s}$, where $\tilde{\gamma}_s$ is the distribution of $\tilde{W}_s$. This relationship reflects the inverse process of $\gamma_t$, and we also have $\tilde{\gamma}_t = \gamma_0$ and $\tilde{\gamma}_0 = \gamma_t$. We can analyze Eq. (39) by using the parametrix method [7]. We refer to Remark 3 for the explanation that Eq. (39) satisfies the assumptions of the parametrix method [7].

Let us express $p_\gamma(y, s|x, s')$ as the transition kernel of Eq. (39). For simplicity, we express the conditional distribution given $\tilde{W}_0 \sim \tilde{\gamma}_0$ as $\tilde{\gamma}_{s=t|s=0}$, which corresponds to the above transition kernel: $\tilde{\gamma}_{s=t|s=0}(y) = p_\gamma(y, s|x = w, s' = 0)$. By fixing $\tilde{W}_0$ as $w$, we obtain $\mathbb{E}_{\gamma_{0|t}}[\|W_0'\|^2|W_t' = w] = \mathbb{E}_{\tilde{\gamma}_{s=t|s=0}}[\|\tilde{W}_t\|^2|\tilde{W}_0 = w]$. By analyzing the reverse process of $\gamma_t$, we can evaluate the second term in the upper bound of $\widetilde{V}_{\nabla_t}$, i.e., $\int \rho_t \mathrm{d}w \|\mathbb{E}_{\gamma_{0|t}}[\nabla F(W_0', B_t')|W_t = w]\|^2$.

We consider approximating $\tilde{\gamma}_{s=t|s=0}$ by the parametrix method to derive the upper bound of $\mathbb{E}_{\tilde{\gamma}_{s=t|s=0}}[\|\tilde{W}_t\|^2|\tilde{W}_0 = w]$. By using the upper bound of the transition kernel provided by the parametrix method in Deck and Kruse [7], we obtain

$$p_\gamma(y, s|x, s') \leq K_1(s - s')^{-d/2}e^{-K_2\frac{\|y-x\|^2}{s-s'}},$$

where $K_1$ and $K_2$ are positive and problem-dependent constants and do not depend on $s - s'$. From the above inequality, by setting $x = w$ and $s - s' = t$, we have

$$\mathbb{E}_{\gamma_{0|t}}[\|W_0\|^2|W_t = w] \leq \tilde{C}_0(\|w\|^2 + \tilde{C}_1 t^2),$$

where $\tilde{C}_0$ and $\tilde{C}_1$ are positive and problem-dependent constants. Thus, we have

$$\mathbb{E}_{\rho_t}\mathbb{E}_{\gamma_{0|t}}[\|\nabla F(W_0', B_t')|W_t = w]\|^2$$
$$\leq 2M^2\tilde{C}_0\left(\tilde{C}_1 t^2 + s^2 + 2\left(1 \vee \frac{1}{m}\right)\left(b + 10M^2\frac{b}{m} + \frac{d}{\beta}\right)\right) + 2M^2\frac{b}{m} =: D_5. \tag{40}$$

In the above, $\tilde{C}_1 t^2$ is negligibly much smaller than the other terms within $0 \leq t \leq \eta$.

From Eq. (38) and Eq. (40), we obtain

$$\widetilde{V}_{\nabla_t} \leq 2(D_4 + D_5) := D_1. \tag{41}$$

We conclude this section by finalizing the proof of Theorem 4. By combining Eqs. (35), (37), and (41) with Eq. (24) and taking the expectation with respect to all of the randomness, we obtain

$$\mathbb{E}_{S,S'}\mathrm{KL}(\rho_\eta|\gamma_\eta) \leq e^{\frac{-\eta}{4\beta c_{\mathrm{LS}}}}\mathbb{E}_{S,S'}\mathrm{KL}(\rho_0|\gamma_0) + (1 - e^{\frac{-\eta}{4\beta c_{\mathrm{LS}}}})D_2$$
$$+ 2c_{\mathrm{LS}}(1 - e^{\frac{-\eta}{4\beta c_{\mathrm{LS}}}})D_3 + 2\beta^2 c_{\mathrm{LS}}(1 - e^{\frac{-\eta}{4\beta c_{\mathrm{LS}}}})D_1,$$

where we used the following fact:

$$\int_0^\eta \mathrm{d}t e^{\frac{t}{4\beta c_{\mathrm{LS}}}} = 4\beta c_{\mathrm{LS}}(e^{\frac{\eta}{4\beta c_{\mathrm{LS}}}} - 1).$$

Since $e^{\frac{-\eta}{4\beta c_{\text{LS}}}} \geq 1 - \frac{\eta}{4\beta c_{\text{LS}}}$ from the assumption, we have

$$\mathbb{E}_{S,S'}\text{KL}(\rho_\eta|\gamma_\eta) \leq e^{\frac{-\eta}{4\beta c_{\text{LS}}}}\mathbb{E}_{S,S'}\text{KL}(\rho_0|\gamma_0) + \frac{\eta}{4\beta c_{\text{LS}}}D_2 + \frac{\eta}{2\beta}D_3 + \frac{\eta\beta}{2}D_1.$$

This concludes the proof.

**Remark 3.** *We show that Eq. (39) satisfies the assumption of the parametrix method [7]. First, Deck and Kruse [7] assumes the strong regularity condition for the diffusion coefficient, which is satisfied because the diffusion coefficient in our setting is a constant. Next, we confirm the assumptions that the drift coefficient $b(w,s) := \nabla F(\tilde{W}_{s=t}, B'_{s=t}) + 2\beta^{-1}\nabla \log \gamma_{t-s}$ must satisfy. Specifically, the following two assumptions for $b(w,s)$ must be satisfied: (i) the locally Hölder continuous condition on some bounded subset in $\mathbb{R}^d$ and (ii) the global growing condition, that is, $\|b(w,s)\| \leq c_0(\|x\|+1)$ with some positive constant $c_0$. Fortunately, for $\nabla F$ in $b(w,s)$, the assumption (i) is satisfied by Assumption 2, and the assumption (ii) holds from Lemma 3. Furthermore, $\nabla \log \gamma_{t-s}$ in $b(w,s)$ also satisfies the assumption (ii) from Lemma E.1 in Mou et al. [24]. According to the fact that $\nabla\gamma_{t-s}$ satisfies the Hölder continuous as shown in [12], we can see that $\frac{1}{\gamma_{t-s}}$ is bounded by considering the bounded set in $\mathbb{R}^d$. This means that $\nabla \log \gamma_{t-s} = \frac{1}{\gamma_{t-s}}\nabla\gamma_{t-s}$ in $b(w,s)$ is Hölder continuous and satisfies the assumption (i).*

### F.3 On relaxing Gaussian condition in Assumption 4

The Gaussian initial distribution assumption for $W_0$ could be relaxed. Let us consider the case when the initial distribution $P_{W_0}$ is a mixture of Gaussian distribution, where each component of $P_{W_0}$ satisfies Assumption 4. In our original proof, the Gaussian assumption is used when deriving the upper bound of the finite second moment at the initial state, and when analytically marginalizing out the initial state of the transition kernel given by the fundamental solution of the parametrix method. Even when using the mixture of Gaussian distribution as the initial distribution, it is possible to satisfy these conditions. The finite second-moment condition can easily be satisfied and the integration of the transition kernel can be executed by focusing on each component of the mixture distribution. Thus, by repeating the similar derivation in Appendices F.1 and F.2, we get the similar upper bound of $\mathbb{E}_{S,S'}\text{KL}(\rho_\eta|\gamma_\eta)$ even when the initial distribution is the Gaussian mixture distribution.

### F.4 Proof of Theorem 5

We first show the proof of Theorem 5.

**Theorem 5.** *Suppose that Assumptions 1 and 2 are satisfied and that $F(w,z)$ is $R$-strongly convex ($0 < R < \infty$). Then, for any $T \in \mathbb{R}_+$, we have*

$$\frac{\partial \text{KL}(\rho_t|\gamma_t)}{\partial t} \leq -\frac{R}{4}\text{KL}(\rho_t|\gamma_t) + \frac{\beta}{2}\mathbb{E}\|\nabla F(W_t, S) - \nabla F(W_t, S')\|^2, \tag{15}$$

*and*

$$|\text{gen}(\mu, P_{W_T|S}; L)| \leq \sqrt{\frac{2\beta\sigma_g^2}{n}\int_0^T e^{-\frac{(T-t)R}{4}}\mathbb{E}\|\nabla F(W_t, S) - \nabla F(W_t, S')\|^2\text{d}t}. \tag{16}$$

*Proof.* Since $f(w,z)$ is $R$-strongly convex function for any $z$, we have

$$\frac{\partial \text{KL}(\rho_t|\gamma_t)}{\partial t} \leq -\frac{1}{2\beta}\mathbb{E}\|\nabla \log \rho_t - \nabla \log \gamma_t\|^2 + \frac{\beta}{2}\mathbb{E}\|\nabla F(W_t, S) - \nabla F(W_t, S')\|^2$$

$$\leq -\frac{R}{4}\text{KL}(\rho_t|\gamma_t) + \frac{\beta}{2}\mathbb{E}\|\nabla F(W_t, S) - \nabla F(W_t, S')\|^2,$$

where we utilized the local LSI in Theorem 5.5.2 of Bakry et al. [2]. Since the stationary distribution si $\pi \propto \exp(-\beta F(x))$, From Theorem 5.5.2 of Bakry et al. [2], $\gamma_t$ satisfies the LSI with the LSI constant $2/(\beta R)$. By integrating $e^{\frac{tR}{4}}\frac{\partial \text{KL}(\rho_t|\gamma_t)}{\partial t}$ over $t \in [0, T]$ and rearranging the above, we obtain the upper bound of $I(W_t; S)$. This concludes the proof. $\qquad\square$

We can obtain the similar result for bounded non-convex losses with $l_2$-regularization $F(w, z) = F_0(w, z) + \frac{\lambda}{2}\|w\|^2$ ($0 < \lambda < \infty$), where $F_0(w, z)$ is $C$-bounded ($0 \le C < \infty$) with the initial distribution $\pi_0 \propto e^{-\frac{\beta\lambda}{2}\|w\|^2}$. From Lemma 34 in Li et al. [20], $\gamma_t$ satisfies the LSI with the constant $\frac{\lambda}{e^{8\beta C}}$. Then, following the way in the proof of Theorem 5, we obtain the bound with replacing $R$ of Eq. (16) to $\frac{\lambda}{e^{8\beta C}}$.

## G   Proofs of generalization analyses directly using a training loss

In this section, we provide our proof for our generalization bounds in the case when the same loss is used for training and the generalization performance evaluation (Corollaries 1 and 2). The key to deriving these bounds is showing that $f$ in SGLD is sub-exponential under Assumptions 2, 3 (Theorem 6). Therefore, we explain how to obtain this result in Appendices G.1 and G.2 before introducing the details of proofs for our bounds in Appendices G.3 and G.4.

### G.1   Preparation for the proof of sub-exponential property

We introduce some auxiliary lemmas that assure the existence of bounded local minima. These are used later for showing the sub-exponential property of a loss function in SGLD.

**Lemma 7.** *Suppose that Assumptions 2 and 3 are satisfied. Then, for each $z \in \mathcal{Z}$, there exists a positive constant $A$ such that*

$$|f(0, z)| \le A.$$

*Proof.* Denote $\tilde{w}_z^*$ as a global minima of $f(\cdot, z)$ for each $z \in \mathcal{Z}$. By using Taylor's theorem around $\tilde{w}_z^*$, for $t \in (0, 1]$, we obtain the following equation with a parameter $\tilde{w}_z = t\tilde{w}_z^*$:

$$f(0, z) = f(\tilde{w}_z^*, z) + \nabla f(\tilde{w}_z^*, z) \cdot \tilde{w}_z^* + \frac{1}{2}\tilde{w}_z^* \cdot \nabla^2 f(\tilde{w}_z, z) \cdot \tilde{w}_z^*.$$

According to Assumption 2 and the fact that $\tilde{w}_z^*$ is the global minima (i.e., $\nabla f(\tilde{w}_z^*, z) = 0$), we obtain

$$f(0, z) \le f(\tilde{w}_z^*, z) + \frac{1}{2}M\|\tilde{w}_z^*\|^2. \tag{42}$$

Farghly and Rebeschini [10] has shown that all the local minima $\tilde{w}_z^*$ are inside the ball in the Euclidean space. That is, for each $z \in \mathcal{Z}$, all $\tilde{w}_z^*$ are located in $\overline{B(0, r)}$ with $r = \sqrt{b/m}$, where $B(x, r)$ ($r > 0$) is the ball in the Euclidean space defined as $B(x, r) := \{x \in \mathbb{R}^d : \|x - y\| < r\}$ and $\overline{B(x, r)}$ is the closure of $B(x, r)$. From this fact, we obtain $\|\tilde{w}_z^*\|^2 \le b/m$ and Eq. (42) can be upper bounded as

$$f(0, z) \le f(\tilde{w}_z^*, z) + \frac{Mb}{2m}.$$

Next, we show that, for each $z \in \mathcal{Z}$, the global minima $f(\tilde{w}_z^*, z)$ is bounded uniformly. Since $f(\tilde{w}_z, z)$ is continuous with respect to $w$ for each $z \in \mathcal{Z}$ under Assumption 2, it is continuous in $C$ when considering the closed set $C := \overline{B(0, r)}$ with $r = \sqrt{b/m}$. From the property of the continuous function in the closed set, the maximum and minimum value of $f(\tilde{w}_z, z)$ is always bounded, i.e., we have $f(\tilde{w}_z^*, z) < \infty$ for each $z \in \mathcal{Z}$. By considering the largest global minimum and denote it as $\tilde{A}$, we obtain $f(\tilde{w}_z^*, z) \le \tilde{A}$ and thus

$$f(0, z) \le \tilde{A} + \frac{Mb}{2m}.$$

This concludes the proof.  □

Under Lemma 7, we can modify the upper and lower bound for $f(w, z)$ in Raginsky et al. [29] as follows.

**Lemma 8** (Modified version from Raginsky et al. [29]). *Suppose that Assumptions 2 and 3 are satisfied. Then, for any $z \in \mathcal{Z}$, we have*

$$\frac{m}{3}\|w\|^2 - \frac{b}{2}\log 3 \le f(w, z) \le \frac{M}{2}\|w\|^2 + M\sqrt{\frac{b}{m}}\|w\| + A.$$

*Proof.* Raginsky et al. [29] assumed that for any $z \in \mathcal{Z}$, there exists constant $A$ such that $\|f(0, z)\| \leq A$. Instead, we show the existence of such $A$ by Lemma 7. On the basis of this fact, we obtain the claim in the same way with Raginsky et al. [29]. □

Finally, we provide the following useful lemma from Farghly and Rebeschini [10].

**Lemma 9** (Lemma B.2 in Farghly and Rebeschini [10]). *Suppose that Assumptions 2, 3 and 4 are satisfied. Then, for any $\eta \in (0, 1 \wedge \frac{m}{5M^2})$, $T \in \mathbb{N}$ and $p \in \mathbb{N}$, we have*

$$\mathbb{E}\|W_T\|_2^{2p} \leq \mathbb{E}\|W_0\|_2^{2p} + c(p), \tag{43}$$

*where*

$$c(p) := \frac{1}{m}\left(\frac{6}{m}\right)^{p-1}\left(1 + \frac{2^{2p}p(2p-1)d}{m\beta}\right)\left[\left(2b + 8\frac{M^2}{m^2}b\right)^p + 1 + 2\left(\frac{d}{\beta}\right)^{p-1}(2p-1)^p\right].$$

If, for any $p \geq 1$, the $L_p$ norm of a random variable $X$ is upper-bounded as $(\mathbb{E}[X^p])^{1/p} \leq C\sqrt{p}$ with some positive constant $C$, then $X$ is a sub-Gaussian random variable (see Proposition 2.5.2 in Vershynin [33]). According to the fact that $p^2 \leq e^{2p}$, it is evident that $W_T$ is a sub-Gaussian random variable from Eq. (43).

## G.2 Sub-exponential property for a loss function

We now provide the complete proof of Theorem 6. We first show the fact that a loss function in SGLD has the sub-exponential property (Appendix G.2) and explain how to evaluate the constants in the sub-exponential condition for deriving our generalization bounds (Appendix G.3).

Recall that the statement of Theorem 6 is as follows.

**Theorem 6.** *Suppose that Assumptions 2, 3 and 4 are satisfied. Let $P_{W_T} = \mathbb{E}_S[P_{W_T|S}]$ be the marginal distribution of the output obtained using the SGLD algorithm at the $T$-th iteration. Then, for any $\eta \in (0, 1 \wedge \frac{m}{5M^2})$ and $T \in \mathbb{N}$, $f(W_T, Z)$ is sub-exponential under the distribution $P_{W_T} \otimes \mu$. That is, there exist positive constants $\sigma_e^2$ and $\nu$ w.r.t. $\{m, \beta, M, b, d, s^2\}$ such that*

$$\log \mathbb{E}_{W_T \otimes Z}\left[e^{\lambda(f(W_T, Z) - \mathbb{E}_{W_T \otimes Z}[f(W_T, Z)])}\right] \leq \frac{\sigma_e^2 \lambda^2}{2} \quad for\ all\ |\lambda| < \frac{1}{\nu}.$$

*Proof.* Now, we proceed to the proof of sub-exponential property. To show the sub-exponential property for $f(w, z)$, it is sufficient to show that there exists a positive number $c_0$ such that $\mathbb{E}e^{\lambda f(w,z)} < \infty$ for all $|\lambda| \leq c_0$ (see Theorem 2.13 in Wainwright [34]). To show this, we follow the proof of Proposition 2.7.1 in Vershynin [33], which uses the Taylor expansion of the exponential moment. By considering the Taylor expansion, we have

$$\mathbb{E}e^{\lambda(f(W_T, Z) - \mathbb{E}f(W_T, Z))} \leq 1 + \mathbb{E}\sum_{p=2}^{\infty}\frac{\lambda^p(f(W_T, Z) - \mathbb{E}f(W_T, Z))^p}{p!}$$

$$\leq 1 + \mathbb{E}\sum_{p=2}^{\infty}\frac{(\lambda e)^p(f(W_T, Z) - \mathbb{E}f(W_T, Z))^p}{p^p}$$

where we used $p! \geq (p/e)^p$, which is obtained by the Stirling's approximation. Later, we restrict the $\lambda$ such that this series converges, and thus, we can swap the sum and expectation. From the fact that $(x + y)^p \leq 2^{p-1}x^p + 2^{p-1}y^p$ for $x, y \geq 0$, we obtain

$$\mathbb{E}(f(W_T, Z) - \mathbb{E}f(W_T, Z))^p \leq 2^{p-1}\mathbb{E}[|f(W_T, Z)|^p] + 2^{p-1}(|-\mathbb{E}[f(W_T, Z)]|)^p.$$

Given the marginal distribution of the parameters obtained by the $T$-th iterate of the SGLD algorithm, i.e., $W_T \sim p(W_T)$, we have the following fact by using the result of Lemma 8 and the Cauchy–Schwartz inequality:

$$\frac{m}{3}\|W_T\|^2 - \frac{b}{2}\log 3 \leq f(W_T, Z) \leq \frac{M}{2}\|W_T\|^2 + M\sqrt{\frac{b}{m}}\|W_T\| + A \leq M\|W_T\|^2 + \frac{b}{2m} + A. \tag{44}$$

By using Eq. (44) and the inequality $(x+y)^p \leq 2^{p-1}x^p + 2^{p-1}y^p$ for $x, y \geq 0$, we have

$$\mathbb{E}f(W_T, Z)^p \leq \mathbb{E}|f(W_T, Z)|^p$$

$$\leq \left( 2^{p-1}M^p\mathbb{E}\|W_T\|_2^{2p} + 2^{p-1}\left(\frac{b}{2m} + A\right)^p \right) \vee \left( 2^{p-1}\left(\frac{m}{3}\right)^p \mathbb{E}\|W_T\|_2^{2p} + 2^{p-1}\left(\frac{b}{2}\log 3\right)^p \right).$$

$$(45)$$

The term $\mathbb{E}\|W_T\|_2^{2p}$ in Eq. (45) can further be upper bounded by utilizing Eq. (43) in Lemma 9. As discussed immediately below Lemma 9, $W_T$ is a sub-Gaussian random variable. Thus, for any $p \in \mathbb{N}$, $(\mathbb{E}[W_T^p])^{1/p} \leq C\sqrt{p}$ holds with some positive constant $C$. Combining Eq. (45) and the above fact leads to

$$\mathbb{E}f(W_T, Z)^p \leq C_0^p + C_1^p p^p,$$

where $C_0$ and $C_1$ are positive constants that only depend on $s^2, m, M, b, d, A$, and $\beta$. For the latter purpose, we introduce $C_5$ as

$$\mathbb{E}f(W_T, Z)^p \leq C_5 p^p,$$

where $C_5$ only depends on $s^2, m, M, b, d, A$ and $\beta$. Then, we have

$$\mathbb{E}e^{\lambda(f(W_T, Z) - \mathbb{E}f(W_T, Z))} \leq 1 + \sum_{p=2}^{\infty} \frac{(\lambda e)^p C_5 p^p}{p^p} = 1 + \sum_{p=2}^{\infty} (\lambda e C_5)^p = 1 + \frac{(\lambda e C_5)^2}{1 - \lambda e C_5},$$

where $\lambda e C_5 < 1$. Moreover, by setting $\lambda e C_5 < 1/2$, we have

$$\mathbb{E}e^{\lambda(f(W_T, Z) - \mathbb{E}f(W_T, Z))} \leq 1 + 2\lambda^2 e^2 C_5^2 \leq e^{2\lambda^2 e^2 C_5^2}.$$

From the above, we can see that $f(W_T, Z)$ is a sub-exponential function with the following constants: $\sigma_e^2 := 4e^2 C_5^2$ and $\nu := \frac{1}{2eC_5}$ where $C_5$ only depends on $s^2, m, M, b, d$ and $A$. $\qquad\square$

### G.3    Proof of generalization error bound directly using a training loss

Here, we provide the complete proof of Corollary 1.

**Corollary 1.** *Suppose that Assumptions 2, 3, and 4 are satisfied. Then, for any $\beta \geq 2/m$, $\eta \in (0, 1 \wedge \frac{m}{5M^2} \wedge 4\beta c_{\mathrm{LS}})$, and $T \in \mathbb{N}$, we obtain*

$$|\mathrm{gen}(\mu, P_{W_T|S}; F)| \leq \Psi^{*-1}\left( \frac{c_1}{n}\left(1 \wedge \frac{\eta T}{4\beta c_{\mathrm{LS}}}\right)(V_\nabla + c_2) \right),$$

*where*

$$\Psi^{*-1}(y) = \begin{cases} \sqrt{2\sigma_e^2 y} & \text{if } y \leq \frac{\sigma_e^2}{2\nu} \\ \nu y + \frac{\sigma_e^2}{2\nu} & \text{otherwise} \end{cases},$$

$c_1$ *and* $c_2$ *are the same as in Theorem 4, and* $\sigma_e^2$ *and* $\nu$ *are the same as in Theorem 6.*

*Proof.* We use the following theorem in Bu et al. [3] to derive the generalization error for sub-exponential losses.

**Theorem 7** (Bu et al. [3]). *Suppose that there exist positive constants $\sigma_e^2$ and $\nu$ such that*

$$\log \mathbb{E}_{W_T \otimes Z}\left[ e^{\lambda(f(W_T, Z) - \mathbb{E}_{W_T \otimes Z}[f(W_T, Z)])} \right] \leq \frac{\sigma_e^2 \lambda^2}{2} \quad \text{for all } |\lambda| < \frac{1}{\nu}.$$

*Then, we have*

$$|\mathrm{gen}(\mu, P_{W_T|S}; F)| \leq \Psi^{*-1}\left( \frac{I(W_T; S)}{n} \right),$$

*where*

$$\Psi^{*-1}(y) = \begin{cases} \sqrt{2\sigma_e^2 y} & \text{if } y \leq \frac{\sigma_e^2}{2\nu} \\ \nu y + \frac{\sigma_e^2}{2\nu} & \text{otherwise.} \end{cases}$$

Substituting the constants of the sub-exponential property shown in Theorem 6 and the upper bound of $I(W_T; S)$ in Theorem 4 into the above completes the proof. $\qquad\square$

### G.4 Proof of an excess risk

We rewrite our corollary as follows.

**Corollary 2.** *Suppose that Assumptions 2, 3, and 4 are satisfied. Then, for any $\beta \geq 2/m$, $\eta \in (0, 1 \wedge \frac{m}{5M^2} \wedge 4\beta c_{\mathrm{LS}})$, and $T \in \mathbb{N}$, we obtain*

$$\mathrm{Excess}(\mu, P_{W_T|S}) = \mathcal{O}\left( \sqrt{\frac{(\eta T \wedge 1)}{n}} + e^{-\eta T/c_{LS}} + \sqrt{\eta} + c_{\mathrm{err}} \right),$$

*where $c_{\mathrm{err}}$ is the positive constant w.r.t. $\{M, m, b, d, \beta\}$ corresponding to the optimization error.*

*Proof.* We can decompose the excess risk at $T$ as

$$\begin{aligned}
\mathrm{Excess}(\mu, P_{W_T|S}) &= \mathbb{E}_{W_T,S}[F_\mu(W_T) - F_S(W_T) + F_S(W_T) - F_\mu(w^*)] \\
&= \mathrm{gen}(\mu, P_{W_T|S}; F) + \mathbb{E}_{W_T,S}[F_S(W_T) - F_\mu(w^*)],
\end{aligned}$$

where the last term is called the *optimization error*. The optimization error can be bounded as

$$\begin{aligned}
\mathbb{E}_{W_T,S}[F_S(W_T) - F_\mu(w^*)] &= \mathbb{E}_{W_T,S}[F_S(W_T) - \min_w F_S(w) + \min_w F_S(w) - F_S(w^*)] \\
&\leq \mathbb{E}_{W_T,S}[F_S(W_T) - \min_w F_S(w)],
\end{aligned}$$

where the above inequality comes from the fact that $\mathbb{E}_{W_T,S}[\min_w F_S(w) - F_S(w^*)] \leq 0$. Let us denote $\epsilon_{\mathrm{opt}}$ as $\mathbb{E}_{W_T,S}[F_S(W_T) - \min_w F_S(w)]$. Then, we can express the upper bound of the excess risk as follows:

$$\mathrm{Excess}(\mu, P_{W_T|S}) \leq |\mathrm{gen}(\mu, P_{W_T|S}; F)| + \epsilon_{\mathrm{opt}}.$$

We first bound the $\epsilon_{\mathrm{opt}}$ term. Using the Gibbs distribution $\pi(\mathrm{d}w) \propto \exp(-\beta F(w, S))$ and the triangle inequality, we obtain

$$\epsilon_{\mathrm{opt}} \leq |\mathbb{E}_{W_T,S}F_S(W_T) - \mathbb{E}_{\pi,S}F_S(W)| + |\mathbb{E}_{\pi,S}F_S(W) - \min_w F_S(w)|, \tag{46}$$

where we express the expectation under the joint distribution $\mu^N \otimes \pi$ as $\mathbb{E}_{\pi,S}$. The first term on the right-hand side of Eq. (46) is the *convergence error* of the SGLD algorithm, which can be seen as $\mathcal{O}(e^{-k\eta/\beta c_{LS}} + \sqrt{\eta})$. From Lemma 6 in Raginsky et al. [29], we have

$$|\mathbb{E}_{W_T,S}F_S(W_T) - \mathbb{E}_{\pi,S}F_S(W_T)| \leq \left( M\sigma + M\sqrt{\frac{b}{m}} \right) \mathbb{E}_S W_2(P_{W_T|S}, \pi),$$

where $\sigma^2 := \mathbb{E}_{P_{W_T|S}}\|W_T\|^2 \vee \mathbb{E}_\pi\|W\|^2$ and $W_2$ is the 2-Wasserstein distance. By using the $T_2$ inequality, we obtain $W_2(P_{W_T|S}, \pi) \leq \sqrt{c_{\mathrm{LS}}\mathrm{KL}(P_{W_T|S}\|\pi)}$. From Theorem 1 in Vempala and Wibisono [32], we further obtain $\mathrm{KL}(P_{W_T|S}\|\pi) \leq \mathcal{O}(e^{-2k\eta/\beta c_{LS}} + \eta)$. Combining these results leads to $|\mathbb{E}_{W_T,S}F_S(W_T) - \mathbb{E}_{\pi,S}F_S(W_T)| = \mathcal{O}(e^{-k\eta/\beta c_{LS}} + \sqrt{\eta})$.

The second term $|\mathbb{E}_{\pi,S}F_S(W) - \min_w F_S(w)|$ corresponds to the *minimization error*, which can be upper-bounded by $c_{\mathrm{err}} := \frac{d}{2\beta} \log\left( \frac{eM}{m} \left( \frac{b\beta}{d} + 1 \right) \right)$ according to Proposition 11 in Raginsky et al. [29]. This completes the proof. $\qquad\square$