# OpenReview forum: "Time-Independent Information-Theoretic Generalization Bounds for SGLD"
_NeurIPS.cc/2023/Conference — NeurIPS 2023 poster_

### Official Review · Reviewer_sqKQ · 2023-06-22

**Soundness:** 3 good
**Presentation:** 3 good
**Contribution:** 3 good
**Rating:** 5
**Confidence:** 2

**Summary:**

The paper provides novel information-theoretic generalization bounds for the SGLD. They derive the generalization error bounds by focusing on the time evolution of the Kullback–Leibler divergence.  The bound is   time-independent and removes the step size dependence in previous works.

**Strengths:**

1. This paper provides novel generalization bounds for SGLD by using the Kullback–Leibler (KL) divergence between the distributions of parameters learned from different training datasets.

2. The authors provide the first information-theoretic generalization error bound and the excess risk bound without surrogate losses.

**Weaknesses:**


1. Although the authors claim they provide the first information-theoretic generalization error bound and the excess risk bound without surrogate losses, they indeed use the dissipativity to fix the non-convexity in the proofs.

2.  I feel a little overclaimed on the stepsize because you use the dissipativity, which can be a kind of weak "strongly convexity". In the strongly convex case of   SGLD,  the stepsize can also be improved.




**Questions:**

see weakness part.

**Limitations:**

Yes

---

> ### Author Rebuttal · Authors · 2023-08-09
>
> We would like to express our deepest appreciation for your thoughtful reviews. We sincerely summarize our responses to you as follows.
>
> ---
> ## For Weaknesses (Questions)
> ### For weakness 1 (about dissipativity):
> The dissipative assumption plays an important role in guaranteeing the geometrical convergence of SGLD to the stationary distribution. We note that convergence to the stationary distribution is crucial for reducing training error since the stationary distribution in this context corresponds to the Gibbs posterior distribution of the given loss function. The dissipative assumption is widely used in the research on sampling or non-convex potential function optimization; thus, it is a fundamental property that enables optimization with SGLD rather than strong constraint conditions for generalization. As discussed by [Mou et al., 2022], the dissipative assumption is weaker than convexity and strong convexity. Many non-convex losses commonly used in practice satisfy this property. For specific examples, please see our response to the question from Reviewer oJ4H.
>
> ### For weakness 2 (about our claim of contributions):
> We believe that we are not overclaiming our contributions.
> As you pointed out, it is indeed possible to improve the convergence rate w.r.t. learning rate if the loss function is strongly convex.
> On the other hand, recent studies on sampling, optimization, and the generalization performance of SGLD, such as [Farghly and Rebeschini, 2021], [Xu et al., 2018], and [Raginsky et al., 2017], have focused on the non-convex losses to understand the deep learning models under the dissipative assumption. Moreover, the information-theoretic generalization analysis of SGLD often focuses on deep learning models whose losses are often non-convex and derives the data and algorithm-dependent generalization error bounds. Following this context, this paper analyzed the generalization error of SGLD dealing with non-convex losses under dissipative assumptions.
>
> Furthermore, the assumptions of our Theorem 4 in Section 3 are similar to those in [Farghly and Rebeschini, 2021], which provided the first time-independent bound for SGLD shown in Theorem 3 in Section 2, except for sub-Gaussianity. There is no need to assume sub-Gaussianity when using the same losses for training and generalization evaluation, enabling a direct comparison between our bound (Corollary 1)  and that of [Farghly and Rebeschini, 2021] (Theorem 3) under the completely same assumptions.
> It is indeed a clear fact that in this fair comparison, our bound avoids the divergence observed in [Farghly and Rebeschini, 2021] due to the decrease in step size. We believe this is not an overclaim but a valid observation.
>
> ## Reference
> Please see our global responses.

---

### Official Review · Reviewer_h7LH · 2023-07-05

**Soundness:** 3 good
**Presentation:** 3 good
**Contribution:** 3 good
**Rating:** 7
**Confidence:** 4

**Summary:**

This paper provides novel generalization bounds for SGLD under smooth and dissipative loss conditions via the information-theoretic approach. The proposed bounds are time-independent and decay to zero as the sample size increases, regardless of the number of iterations and whether the step size is fixed. In addition, the proposed bound also leads to an improved excess risk bound by combining the analysis with the existing non-convex optimization error bounds.

**Strengths:**

The combination of information-theoretic bounds and Fokker–Planck (FP) equation establishes the tighter bounds provided in the paper, which is impressive.

**Weaknesses:**

I did not note any significant weakness in the paper.

**Questions:**

It is shown in [1] that the generalization error of the Gibbs algorithm equals the symmetrized KL information between W and S. Under the sub-Gaussian condition, it can be further proved that the generalization error of the Gibbs algorithm has the order of $O(1/n)$ independent of dimension d and time t.

As shown in Table 1, it seems that the bound provided by Farghly and Rebeschini is able to recover this 1/n rate for the generalization error Gibbs distribution but with an extra term $\eta^{1/2}$ for SGLD.

However, the bound provided in this paper only scales in the order of $O(1/\sqrt{n})$. Could you please provide more insights into whether this gap is due to the mismatch between the Gibbs distribution and the SGLD or it is due to the looseness of mutual information-based bound? My guess would be the issue of mutual information-based bound, and maybe the techniques developed in the following recent paper [2,3] might be helpful.

[1] Aminian, Gholamali, Yuheng Bu, Laura Toni, Miguel Rodrigues, and Gregory Wornell. "An exact characterization of the generalization error for the Gibbs algorithm." Advances in Neural Information Processing Systems 34 (2021): 8106-8118.

[2] Zhou, Ruida, Chao Tian, and Tie Liu. "Exactly Tight Information-Theoretic Generalization Error Bound for the Quadratic Gaussian Problem." arXiv preprint arXiv:2305.00876 (2023).

[3] Wu, Xuetong, Jonathan H. Manton, Uwe Aickelin, and Jingge Zhu. "Fast rate generalization error bounds: Variations on a theme." In 2022 IEEE Information Theory Workshop (ITW), pp. 43-48. IEEE, 2022.



**Limitations:**

The limitations of the proposed method have been addressed well in the last section of the paper.

---

> ### Author Rebuttal · Authors · 2023-08-09
>
> We would like to express our deepest appreciation for your insightful reviews and for giving us very convincing papers.
> We sincerely summarize our responses to you as follows.
>
> ---
>
> ## For Questions
> Thank you for your insightful feedback.
> As you pointed out, our derived bounds are based on the mutual information between $S$ and $W$, which often leads to slow convergence with the order O(1/sqrt{n}), and thus, our bound inherits same slow convergence order (please see [Bu et al., 2020] and [2,3] you raised.)
> It might indeed be possible to derive improved bounds on the rates if the techniques presented in [2,3] you provided can be utilized in a generalization error analysis for SGLD.
>
> Since the objective of this paper is to derive the time-independent information-theoretic bound in the SGLD context, our analysis was conducted under the most fundamental bound in Theorem 1. We believe that improving the convergence rate is an essential task for the future.
>
> ## Reference
> Please see our global responses.

---

### Official Review · Reviewer_oJ4H · 2023-07-05

**Soundness:** 2 fair
**Presentation:** 1 poor
**Contribution:** 2 fair
**Rating:** 6
**Confidence:** 2

**Summary:**

The paper develops "time-independent" information-theoretic generalization bounds for SGLD. The key ingredients are: assuming the smoothness and dissipativity assumptions of the loss function , tracking evolution of the weight densities for two independent training sets, and their KL divergence. Such an approach avoids accumulation of the mutual information quantities over iterations.

**Strengths:**

The overall approach taken appears correct, and the results, if correct, are interesting and novel. It is unfortunate that the paper is not written clear enough for me assess the correctness of the results.

**Weaknesses:**

I think some important aspects of the paper are not presented clearly or contain errors.

1. The definition of dissipativity appears wrong in Assumption 3: the right-hand side of the inequality is a not a scalar. Although this is easily fixable, it is upsetting to this reviewer that a central concept of the paper is not treated with care.
2. In section 4 when the authors try to develop generalization bounds of SGLD without a surrogate loss. It is not clear to me how one would perform SGLD on say, the 0-1 loss, instead of on a differentiable surrogate loss: How would you get a gradient signal? -- In the opening paragraph of section 4, the paper states $l=f$. I speculate that the authors may be considering the case where SGLD is run on a differentiable surrogate loss but the generalization error is evaluated using the true loss (say 0-1 loss). If that is the case, I would imagine that somewhere a relationship between the true loss and the surrogate loss be given or assumed. But nowhere I can find it. That confused me and stopped me from further decoding what the authors intend to mean.

[After Rebuttal] I have read the authors' rebuttal and raised my rating to Weak Accept.

**Questions:**

I am not familiar with dissapative losses. Assuming dissipativity is defined properly, are dissipative losses practically relevant? Can the authors give some loss functions that we use in practice satisfying this condition?

---

> ### Author Rebuttal · Authors · 2023-08-09
>
> We would like to express our deepest appreciation for your time and careful check.
>
> ---
> ## For Weaknesses
> ### For weakness 1 (carelessness for dissipativity):
> First and foremost, we would like to extend our sincerest apologies for our carelessness.
> As you pointed out, there was a typo in the dissipative assumption. Although the right-hand term in Definition 3 should be $\nabla f(w,z) \cdot w$, we accidentally dropped $\nabla$ in front of $f(w,z)$. To be precise, the definition of dissipativity should be: $m||w||^2-b\leq \nabla f(w,z)\cdot w$. We apologize and have corrected it here.
>
> ### For weakness 2 (about the explanation in Section 4):
> We sincerely regret our confusing explanations.
> We would like to respectfully summarize and explain our intentions and contributions to you.
>
> Section 4 considers a scenario where we directly evaluate the generalization performance using a differentiable loss function $f$ (such as squared loss or logistic loss) employed during training. In other words, we examine the case where the loss function used for evaluating generalization performance ($l$) is the same as the loss function used during training ($f$). We thus do not consider the case when the non-differentiable loss such as the $0$-$1$ loss is used for training (e.g., we **do not assume setting 0-1 loss as $f$** as you mentioned). We regret that expressions such as *without surrogate losses* or *$l=f$* caused confusion. We have removed these expressions from our paper.
>
> The case in which losses used for training are also used for performance evaluation is commonly employed in the theoretical studies of sampling and optimization for Langevin dynamics (please see [Raginsky et al., 2017] and [Xu et al., 2018]). In such studies, theoretical analyses are conducted based on the dissipative assumption that can be handled many non-convex losses, with a focus on deep learning. While the convergence of training errors or speed of convergence to a stationary distribution has been well-analyzed by the above studies, theoretical analysis of generalization ability is still insufficient. This background motivates us to conduct information-theoretic generalized error analysis, which also focuses on deep learning models, in a setting where the loss functions are identical for training and performance evaluation. In essence, our purpose is to enhance our understanding of both the optimization and generalization aspects of SGLD by analyzing generalization ability under the same settings and assumptions utilized in sampling and optimization research.
>
> Why couldn't we perform a generalization error analysis based on the information-theoretic approach in the above setup? This is because the tail behavior of the loss function distribution used for training is unknown under the dissipative assumption (here, the loss function distribution is a distribution w.r.t. the dataset); thus, it is unclear whether the sub-Gaussian assumption is appropriate to assume for the dissipative losses.
> This contrasts the settings in Section 3, where alternative losses with known tail behaviors were chosen (e.g., $f$ being a differentiable loss and $l$ being a 0-1 loss) to perform information-theoretic generalization error analysis for SGLD.
> Our contribution is to overcome this situation by showing the loss function with dissipativity and smoothness has the sub-exponential property in SGLD.
> This allows us to apply the information-theoretic generalized error analysis to the dissipative and smooth losses commonly used in sampling and optimization studies of SGLD.
>
> ## For Questions
> The dissipative losses include all strongly convex and convex losses. The dissipative losses also include losses that are strongly convex or convex when sufficiently far from zero, that is, there exists $\{m, R\}>0$ such that $\forall x, y \ \ \mathrm{s.t.} \ \ ||x-y||^2 \geq R, \ \  (x-y) \cdot (\nabla f(x,z) - \nabla f(y,z)) \geq m||x-y||^2$ or $\forall x, y \ \ \mathrm{s.t.} \ \ ||x-y||^2 \geq R, \ \ (x-y)\cdot (\nabla f(x,z) - \nabla f(y,z)) \geq 0$ (please see [Mou et al., 2022]).
> This means that the dissipative losses include the non-convex losses that have a local optimum somewhat close to zero and losses whose tail behavior is similar to the strongly convex losses. A typical example of losses that satisfy dissipativity is non-convex losses with $l_{2}$ regularization (please see [Mou et al., 2018]). Because of its capability to handle many non-convex losses, the dissipative condition is often employed in the theoretical analysis of non-convex optimization, such as [Raginsky et al., 2017] and [Xu et al., 2018].
>
> When considering the negative log-likelihood losses, which are often used in the Bayesian context, the likelihood distribution that satisfies the Poincar\'{e} inequality exhibits the dissipative property (please see [Bakry et al., 2013] and [Vempara and Wibisono, 2019]).
> Poincar\'{e} inequality is applicable to a wide range of practical likelihood distributions, such as log-concave distributions, distributions obtained via bounded perturbations of Poincar\'{e}-inequality-satisfying (PIS) distributions, distributions with Gaussian convolution added to bounded losses, distributions formed by Lipschitz continuous transformations of PIS distributions, and direct sums of PIS distributions [Bakry et al., 2013]. Thus, this includes many useful models, including deep learning models [Mou et al. 2022; Vempara and Wibisono, 2019].
>
> In essence, the dissipative assumption allows for the broad treatment of not only general non-convex optimization in machine learning but also non-convex losses used in Bayesian inference and Bayesian machine learning.
> On the other hand, it is essential to note that thick-tailed losses, such as long-tailed t-distributions or Cauchy distributions, cannot be handled by the dissipativity assumption [Mou et al. 2022].
>
> ## References & modifications
> Please see our global responses.

---

> > ### Comment · Reviewer_oJ4H · 2023-08-13
> > **Thank you for your clarification**
> >
> > I now understand the setting for Section 4 and appreciate the clarification on dissipative losses.
> >
> > I will raise my rating to above the acceptance threshold.
> >
> > But regarding the error on the definition of dissipativity, I still find it upsetting. I would like to advise the authors that such an error (i.e, one on a central concept or result) should never occur in a submitted technical document by a professional. As a reviewer, I would think: if the authors are so sloppy on the key definitions, how could I trust their derivations that are more technical and might require subtle considerations? -- The reviewers are volunteers, spending their own time trying to understand your paper. If you do not demonstrate a professional attitude and take your own paper seriously, on what ground would you expect the reviewers to be serious about your work?

---

> > > ### Author Response · Authors · 2023-08-15
> > > **Sincere Apologies**
> > >
> > > Dear Reviewer oJ4H,
> > >
> > > We would like to express our genuine gratitude for taking the time to confirm our paper and our response and for providing valuable feedback.
> > > We are pleased that the clarification we provided in our response helped you better understand the setting.
> > > Your willingness to reconsider the score is also deeply appreciated.
> > >
> > > We wholeheartedly understand your frustration and deeply regret any disappointment or concern the error in the definition of dissipativity may have caused. You are absolutely right in highlighting that errors in fundamental concepts should never be present in a professional technical document. We would like to assure you that we have sincerely rectified this error.
> > >
> > > We understand the significance of your concerns regarding the potential impact of such inadvertent errors on the credibility of our derivations and technical explanations.
> > > We are committed to continually ensuring that our professionalism and attention to detail are evident in our present and future endeavors.
> > >
> > > We also wish to emphasize our profound appreciation for your time and effort in comprehending and reviewing our work.
> > > In responding to this dedication, we are keenly aware of our responsibility to meet the standards you rightfully expect.
> > >
> > > Once again, please accept our sincere apologies for any frustration or doubts caused by our error. We are truly grateful for your patience and understanding.
> > >
> > > Sincerely,
> > > Authors.

---

### Official Review · Reviewer_7RWp · 2023-07-06

**Soundness:** 3 good
**Presentation:** 3 good
**Contribution:** 3 good
**Rating:** 6
**Confidence:** 3

**Summary:**

In the literature, there are two approaches to obtain generalization performance bounds. One approach is to study the algorithmic stability and there are well-known results that can bound the generalization error using the algorithmic stability bounds. Another approach is the information-theoretic analysis, which is the approach adopted in this paper. One major contribution of the paper is that it provides generalization bounds that do not grow linearly in time, and moreover, it also avoids an unnatural dependence on the stepsize. In addition, the paper provides additional results that provide information-theoretic generalization bound and the excess risk bound without relying on the surrogate losses. The author(s) are very familiar with the literature and the comparisons with the existing literature is ample and the contributions of the paper are clear.

**Strengths:**

One major contribution of the paper is that it provides generalization bounds that do not grow linearly in time, and moreover, it also avoids an unnatural dependence on the stepsize. In addition, the paper provides additional results that provide information-theoretic generalization bound and the excess risk bound without relying on the surrogate losses. The author(s) are very familiar with the literature and the comparisons with the existing literature is ample and the contributions of the paper are clear.

**Weaknesses:**

One weakness is that even though the paper claims that the bound is independent of time $T$, I think some additional analysis is needed in Theorem 4 to make it more clear. For example, in Theorem 4, there is the term $V_{\nabla}:=\max_{t\in[0,T]}\beta\text{Var}[\nabla F(W_t,B_t)]$. I think the paper should include a lemma or something like that in the Appendix to explain how to provide an upper bound on $V_{\nabla}$ that is uniform in $T$. My guess is that Lemma 6 in the Appendix or something like that can help. But it is better to provide an upper bound on $V_{\nabla}$ to make it more clear your bound is uniform in $T$.

**Questions:**

(1)	In the last sentence of Theorem 1, ``an training dataset’’, here ``an’’ should be ``a’’.
(2)	In the paragraph before Theorem 1, can you provide a reference to the claim that ``This assumption is necessary to obtain the information-theoretic generalization bound.’’?
(3)	Assumption 4 seems to be a bit restrictive. Is it possible for example to relax this assumption to mixture of Gaussians or relax it in some other directions?
(4)	In eqution (10), use \left( and \right).
(5)	Page 6. ``By integratng Eq. (12) over $t\in[0,\eta]$, we have’’ This does not seem to be rigorous. It seems to be that you should consider $\partial/\partial t(KL*e^{\frac{1}{2\beta c_{LS}}t})$ in (12) instead and then integrate over it.
(6)	In the statements of Corollary 1 and Corollary 2, ``holds’’ should be ``hold’’.
(7)	In remark 4, can you double check it is $s^2>1$? It seems to me it should be $s^2<1$ instead.
(8)	In Remark 3 ``we often assume’’. It is better to write ``It is often assumed’’.
(9)	In equation (19) on page 14, what is the integral $\int\left(\frac{\rho_{t}}{\gamma_{t}}\frac{\partial\gamma_{t}}{\partial t}\right)$ w.r.t.? Did you miss $dw$ or what? Please double check the other integrals on page 14 and page 15 as well.
(10)	At the end of the first paragraph on page 17, what is $\mathcal{W}[A]$? What does $[A]$ stand for?
(11)	In the equation before equation (28), what does the notation $w^{d}$ stand for?


**Limitations:**

The authors have adequately discussed the limitations.

---

> ### Author Rebuttal · Authors · 2023-08-09
>
> We would like to express our deepest appreciation for your insightful reviews.
> We sincerely summarize our responses to you as follows.
>
> ---
> ## For Weaknesses
> We appreciate your careful check. As you pointed out, the term $V_{\Delta}:=\max_{t\in\[0, T\]}\beta \mathrm{Var}[\nabla F(W_{t}, B_{t})]$ can be bounded with independent to $T$ and $\eta$ by decomposing the variance at each time step into the sum of the gradient norm and applying Lemma 6. We have newly created Appendix C.3 and supplemented the explanation by adding the lemma. Then, we modified the following sentence before line 167:
> - ``In Appendix C, we provide further discussion, including the derivation of the upper bound of $\beta \mathrm{Var}[\nabla F(W_{t}, B_{t})]$ with independence concerning $T$ and $\eta$.’’
> ---
> ## For Questions
> ### For questions (1), (4), (6), (8) (mainly focusing on typos):
> We sincerely apologize for any inconvenience caused due to some typos.
> We have fixed the above and carefully checked the other statements and equations overall paper.
>
> ### For question (2) (about the citation before Theorem 1.):
> We have modified the sentence, ``This assumption is…,’’ in lines 95–96 as follows ([24] == [Russo and Zou, 2016]; [31] == [Xu and Raginsky, 2017]):
> - ``Assumptions regarding the tail behavior of the loss function distributions, such as sub-Gaussianity, have been adopted in several studies, including Russo and Zou [24] and Xu and Raginsky [31], to obtain information-theoretic generalization bounds.’’
>
> The issue with our original explanation is that it gives the impression that sub-Gaussian is a necessary assumption for deriving the information-theoretic generalization bound. Certainly, an assumption for loss functions w.r.t. randomness of datasets is necessary to derive the information-theoretic generalization bounds; however, that does not imply that sub-Gaussian is always necessary, as you know. For example, the sub-exponential, sub-gamma, or Lipschitzness assumptions are also widely used (please see [Russo and Zou, 2016]). We deeply regret any confusion that our misleading statements may have caused.
>
> ### For question (3) (about Assumption 4):
> Following your advice, we have reconsidered relaxing Assumption 4.
> We then found that Theorem 4 holds even when using a mixture distribution with each Gaussian distribution having a variance $s^{2}<1$.
> This can be done by evaluating KL divergence in the same way as Theorem 4 for each component of the mixture Gaussian distribution thanks to the property of the convexity of the KL divergence.
> We believe these discussions are too complex to be presented in the main part of this paper.
> Therefore, we have created Assumption C.4 and summarized the above explanation. We have further added the following sentence in the footnote on Assumption 4:
> - ``The Gaussian assumption can be relaxed, e.g., to Gaussian mixture, in the theorems and corollaries shown in this paper. The detailed discussions are provided in Appendix C.4.’’
>
> ### For question (5) (about the integral in Eq.(12)):
> You are right.
> We have fixed ``By integrating Eq. (12)...’’ in line 214 as:
> - ``By integrating $e^{\frac{t}{2\beta c_{\mathrm{LS}}}} \frac{\partial \mathrm{KL}(\rho_{t}|\gamma_{t})}{\partial t}$ in (12) over $t\in[0,\eta]$ and rearranging it, we obtain…’’
>
> ### For question (7) (about $s^{2}>1$):
> Thank you for carefully checking our manuscript.
> We have modified it as $s^2<1$ in Remark 4.
> As you are concerned, the condition $s^{2} < 1$ is necessary to ensure that the Gaussian integral does not diverge and to satisfy the initial condition of Corollary 1.
>
> ### For question (9) (about the integral in Eq. (19)):
> We deeply appreciate for reviewing the appendix part of our paper.
> As you pointed out, we evaluate the integral in Eq.(19) w.r.t. dw.
> We apologize and have made the corrections in lines 486–492.
>
> ### For question (10) (about W[A] in p.17):
> The term [A] is a typo. We have deleted this.
> We are sorry for the inconvenience.
>
> ### For question (11) (about $\mathrm{d}w^{d}$):
> We expressed the integral w.r.t. each $d$-dimensional parameters through $w^{d}$.
> However, it should have been written $\mathrm{d}w$ for consistency with other descriptions.
> We have modified the term $\mathrm{d}w^{d}$ in line 548–555 as $\mathrm{d}w$.
>
> ## Reference
> Please see our global responses.

---

### Author Rebuttal · Authors · 2023-08-09

Dear all PCs, SACs, ACs, and reviewers,

We sincerely appreciate your consideration of our paper.
We also would like to extend our deepest appreciation to the reviewers for providing insightful reviews.
Before summarizing our responses to each of the reviewers, due to word limitation, we organize the references and the modified sentences based on the received reviews as follows.
We would greatly appreciate it if you could confirm this.

Best regards,
Authors

---
## References
- [Russo and Zou, 2016]: D. Russo and J. Zou. Controlling bias in adaptive data analysis using information theory. In Proceedings of the 19th International Conference on Artificial Intelligence and Statistics, volume 51, pages 1232–1240, 2016. http://proceedings.mlr.press/v51/russo16.pdf
- [Xu and Raginsky, 2017]: A. Xu and M. Raginsky. ``Information-theoretic analysis of generalization capability of learning algorithms.’’ In Advances in Neural Information Processing Systems, volume 30, pages 2524–2533, 2017. https://arxiv.org/pdf/1705.07809.pdf
- [Raginsky et al., 2017]: M. Raginsky, A. Rakhlin, and M. Telgarsky. ``Non-convex learning via stochastic gradient Langevin dynamics: A nonasymptotic analysis.’’ In Proceedings of the 2017 Conference on Learning Theory, volume 65, pages 1674–1703, 2017. https://arxiv.org/pdf/1702.03849.pdf (corresponds to [23])
- [Xu et al., 2018]: P. Xu, J. Chen, D. Zou, and Q. Gu. ``Global convergence of Langevin dynamics based algorithms for nonconvex optimization.’’ In Advances in Neural Information Processing Systems, volume 31, pages 3126–3137, 2018. https://arxiv.org/pdf/1707.06618.pdf (corresponds to [32])
- [Mou et al., 2022]: W. Mou, N. Flammarion, M. J. Wainwright, and P. L. Bartlett. ``Improved bounds for discretization of Langevin diffusions: Near-optimal rates without convexity.’’ Bernoulli 28(3): 1577–1601, August 2022. https://doi.org/10.3150/21-BEJ1343. https://projecteuclid.org/journals/bernoulli/volume-28/issue-3/Improved-bounds-for-discretization-of-Langevin-diffusions--Near-optimal/10.3150/21-BEJ1343.short (corresponds to [18])
- [Bakry et al., 2013]: D. Bakry, I. Gentil, and M. Ledoux. ``Analysis and Geometry of Markov Diffusion Operators,’’ volume 348. Springer Science & Business Media, 2013. https://link.springer.com/book/10.1007/978-3-319-00227-9 (corresponds to [1])
- [Vempara and Wibisono, 2019]: S. Vempala and A. Wibisono. ``Rapid convergence of the unadjusted Langevin algorithm: Isoperimetry suffices.’’ Advances in neural information processing systems, 32:8094–8106, 2019. https://arxiv.org/pdf/1903.08568.pdf (corresponds to [25])
- [Bu et al., 2020]: Y. Bu, S. Zou, and V. V. Veeravalli. ``Tightening mutual information-based bounds on generalization error.’’ IEEE Journal on Selected Areas in Information Theory, 1(1):121–130, 2020. https://arxiv.org/pdf/1901.04609.pdf
- [Farghly and Rebeschini, 2021]: T. Farghly and P. Rebeschini. ``Time-independent generalization bounds for SGLD in non-convex settings.’’ In Advances in Neural Information Processing Systems, volume 34, pages 19836–19846, 2021. https://openreview.net/pdf?id=tNT4APQ0Wgj
---
## Modifications
### From 7RWp's review
(We show them in our response for Reviewer 7RWp)
### From oJ4H's review
The following additions and corrections have been made to clearly explain the relationship between $f$ and $l$ and dissipativity.
- in lines 9–11: ``Additionally,...when the same loss is adopted for training and the generalization performance evaluation by showing…’’
- in lines 58–59: ``Another contribution is…when the same loss is used for training and the generalization performance evaluation.’’
- in lines 118–122: ``Another limitation of the information-theoretic approach appears in the setting where training losses ($f$) are also used for performance evaluation ($l$), which is often employed in sampling and non-convex optimization studies of SGLD [23, 32]. We could not conduct the information-theoretic analysis in this setting because the tail behavior of the distribution of the training loss, such as Assumption 1, is unclear.’’
- in line 123 (paragraph title): ``Time-independent generalization bounds under the same loss for training and the performance evaluation.’’
- in lines 123–126: ``To solve the above problems, Farghly and Rebeschini…when the same loss is adopted in training and performance metric from the stability…’’
- in lines 144–145: ``Specifically, the proposed bounds are…, and are available in both scenarios where the same loss is used for both training and generalization performance evaluation and when different losses are used.’’
- Title of Section 4. ``Generalization analysis for SGLD directly using a training loss’’
- in lines 227–229: ``Hereinafter, we consider the setting that the generalization performance is measured by a training loss $f$ directly.’’
- in lines 236–237: ``However, when using a training loss directly for the generalization performance measure, …’’
- Subsection title of Section 4.2: ``Generalization bounds for SGLD using the same loss for training and evaluation’’
- in line 293 (paragraph title): ``SGLD analysis with/without changing losses.’’
- in the statement of Table 1: ``The symbol * means…when using the same loss for training and evaluation of generalization performance.’’
- Section title of Appendix D.: ``Proofs of generalization analyses directly using a training loss’’
- in lines 573–574: ``In this section, we…in the case when the same loss is used for training and the generalization performance evaluation….’’
- Sub-section title of Appendix D.4: ``Proof of generalization error bound directly using a training loss’’
- the footnote of Assumption 4.: ``This assumption holds not only for (strongly) convex losses but also for many practically used non-convex loss functions [18]. For example, it applies to non-convex loss functions with $l_{2}$ constraints and likelihood functions that satisfy Poincar\'{e} inequality, such as Gaussian negative log-likelihood [1, 25].’’

---

### Comment · Area_Chair_vJFH · 2023-08-13
**Further clarifications from the AC**

Dear authors, I would like for some further clarifications that would help me, AC, be more focused in the discussion.

I would like to reiterate and restate a concerns that was raised by Reviewer oJ4H.

Can you provide a concrete example (even toy example) of a dissipative function, where the aforementioned bound yields a non trivial generalization bound?

I would like to consider the following two points which makes me wonder if such an example might exist

- As shown by [SSSS'08], there exists a convex problem where the unique solution of the ERM fails to generalize. As such, any optimization algorithm that is ran for infinite horizon must do one of two: Either it never converges to the minimum, or it fails to generalize. Taken with your result I am guessing that the SGLD algorithm will just fail to converge to the minimum even after infinite many iterations, which makes me wonder if it will have any guarantees on the population loss.

- It was also recently showsn [L'22] that every algorithm that guarantees non-trivial population loss on convex problems, must carry dimension-dependent information on the sample. Together with your Thm4, that means that if the temperature is dimension independent, then SGLD will not achieve non-trivial population loss on the (convex) construction in [L'22]. Alternatively, one could choose dimension dependent temperature in SGLD but then algorithmic-independent generalization bounds can be easily (and have been) obtained via standard covering number argument/ uniform convergence. In that case, then, the fact that the generalization doesn't depend on the horizon is insignificant.

Both points do not contradict your result, but they do make me wonder if you could actually provide a concrete example where the aforementioned techniques yield an interesting bound.


SSS'08 -- Stochastic Convex Optimization
L'22 -- Information theoretic lower bounds for information theoretic upper bounds.

---

> ### Author Response · Authors · 2023-08-15
> **Acknowledgements**
>
> Dear Area Chair vJFH,
>
> We appreciate you taking the time to consider our paper and raising very interesting questions in the context of the information-theoretic analysis under the dissipative loss assumption. We have read the references you provided and would like to carefully consider your questions before answering. We would greatly appreciate it if you could grant us about two days to send our response, given that we understand our discussion time is limited. Thank you very much for your understanding.
>
> Sincerely,
> Authors

---

> ### Author Response · Authors · 2023-08-18
> **Author responses (1)**
>
> First and foremost, we would like to express our gratitude for the intriguing questions regarding our paper and the information-theoretic (IT) analysis based on the dissipative assumption. In our opinion, the concerns related to the convergence of the generalization error bounds mentioned, as well as the concerns regarding dimension dependence, are fundamental drawbacks within the context of IT-based generalization analysis for SGLD when dealing with (non-)convex loss functions. We believe these are significant issues that warrant further discussion within the entire research community. Therefore, it might be challenging to provide examples that demonstrate non-trivial upper bounds capable of addressing these concerns based on the dissipativity assumption; however,  we would like to provide an example of a dissipative loss function such that our approach is at least useful for analyzing the generalization error.
>
> Our response consists of the following four elements: ***(i) Background of introducing the dissipativity assumption***, ***(ii) Difference in motivation between bounds based on IT analysis and bounds for uniform convergence (including our contribution)***, ***(iii) Relevance to [sss'08] and [L'22]***, and ***(iv) Specific examples demonstrating the effectiveness of our approach (IT-based analysis with dissipative assumption)***.
>
> ---
> ## Part (i): Background of introducing the dissipativity assumption
>
> SGLD updates the model parameters using the stochastic gradient, which is utilized in SGD, multiplied by the learning rate, and augmented with added Gaussian noise.
> Introducing Gaussian noise to the stochastic gradient allows for escaping local minima and facilitates global parameter exploration. SGLD has garnered attention as a means for theoretically analyzing the performance of models with non-convex losses, such as neural networks (NN), which are commonly used in recent machine learning applications.
> Given that NN models are often trained in SGD, SGLD is a promising theoretical tool due to its similarity to SGD and its ability to perform global parameter exploration in the non-convex optimization context. When conducting theoretical analysis on models with non-convex losses based on the SGLD algorithm, ensuring ergodicity is essential to guarantee convergence to the target distribution and to the (local) minima.
>
> To assure ergodicity for a non-convex potential function, the concept of dissipativity is introduced, relaxing the assumption of convexity. In other words, the dissipative assumption is considered to enable theoretical analysis centered on non-convex losses through the SGLD algorithm. Since convex losses inherently satisfy dissipativity, employing this assumption in the realm of convex analysis may not yield non-trivial results, such as addressing your concern about convergence to the minima or the dimension dependence concerns in SGLD.
>
> However, at least within the context of IT-based analysis, we have discovered that our proposed analysis with dissipativity can yield intriguing results even when dealing with convex losses. The details will be explained in Part (iii).

---

> > ### Author Response · Authors · 2023-08-18
> > **Author responses (2)**
> >
> > ## Part (ii): Difference in motivation between bounds based on IT analysis and bounds for uniform convergence (including our contribution)
> >
> > In order to effectively address your concerns from [SSS'08, L'22], we would like to explain the differences in motivation between IT analysis and generalization analysis based on uniform convergence as well as our contributions within this context.
> >
> > ---
> > ### Part (ii)-1: Differences of each bound
> > The generalization error bound based on the notion of uniform convergence (UC) is established from the perspective of the worst-case scenario regarding the generalization performance offered by the trained model. This bound guarantees, as you know, that the generalization error of all hypotheses in the algorithm’s output space simultaneously vanishes as the size of the training data increases, ensuring the convergence of generalization error. Furthermore, this bound asserts that within the ERM principle, it suffices to output any hypothesis from the class that minimizes empirical risk, and by measuring model complexities such as VC dimension or Rademacher complexity, one can evaluate generalization performance. In other words, this bound offers non-trivial guarantees when the hypothesis class utilized by the algorithm, along with its complexity, is moderately constrained.
> >
> > On the other hand, deep learning models are included in vast hypothesis classes where the model complexity drastically increases with the size and depth of the network. When applied to such models, UC-based bounds turn into a vacuous metric due to the exceedingly large model complexity.
> >
> > Furthermore, bounds based on UC rely solely on the hypothesis space and are unable to leverage beneficial statistics obtained from algorithms or datasets, which sometimes results in an inability to capture the true essence of generalization performance. For instance, the gradient variance w.r.t. model parameters exhibit a strong correlation with the generalization performance of deep learning [Jiang et al., 2020]; however, this correlation cannot be represented within UC-type bounds. (We refer [Amit et al., 2022] for details.)
> > This observation leads to the recent interest in data and algorithmic-dependent generalization error bounds, such as the PAC-Bayes and IT generalization error bounds.
> >
> > The strength of the IT-based analysis lies in its capacity to directly incorporate the algorithm- and data-dependent statistics related to the generalization performance, such as the gradient variance instead of the model complexity, into the generalization error upper bounds. Especially, the gradient variance is empirically known to have a stronger correlation with the generalization performance of NNs [Jiang et al., 2020] in comparison to statistics appearing in uniform convergence analysis contexts (e.g., VC dimension, the number of parameters $d$, and the norm of parameters).
> > Therefore, the gradient variance has been an increasingly significant metric in recent generalization analyses.
> >
> > Although the gradient variance implicitly depends on the dimension of parameters $d$, it is widely recognized that, in practice, the gradient variance becomes reasonably small as the training proceeds [Jiang et al., 2020]. Consequently, IT-based bounds offer a sensible generalization error bound even in models with significantly high complexity, such as NN models. This is why it has gained attention in the context of SGLD's generalization analysis.
> >
> > In short, the core aim of IT-based analysis is to offer practical bounds that effectively account for models with high complexity, like NN models, by directly integrating empirically validated statistics associated with generalization obtained from datasets and an algorithm.
> > Active discussions within the realm of IT-based analysis revolve around how to analyze the generalization performance of NN, which involve non-convex losses, to derive bounds that lead to an accurate understanding of generalization performance.

---

> > > ### Author Response · Authors · 2023-08-18
> > > **Author responses (3)**
> > >
> > > ### Part (ii)-2: Our contributions
> > > Our study follows this context and proposes a theoretical analysis method to provide a more accurate understanding and evaluation of generalization bounds while taking advantage of the strengths of IT-based analysis.
> > >
> > > Existing IT-based generalization error bounds expressed with the gradient variance have a structure that involves summing up gradient variances across all time steps, leading to divergence unless sufficiently decreasing the step size with $t$ (please see Theorem 2 of our paper). Additionally, it's worth highlighting that the gradient variance at the initial time step doesn't exhibit a significant correlation with generalization, as noted by [Jiang et al., 2020]. This observation implies that uniformly treating the value of gradient variance at each step might not be a practical approach.
> > >
> > > Against this backdrop, we have conjectured that we could achieve a more practically relevant analysis of generalization performance by avoiding summing up gradient variances equally across all time steps.
> > > Under this hypothesis, we focused on how the KL-divergence of ​​two different parameter distributions obtained by the SGLD algorithm changes under the different training datasets instead of recursively analyzing the MI across all steps (please see Lemma 1 in our paper).
> > > As shown in Eq. (9),  thanks to the LSI inequality, the contribution of the gradient variance at the early time step to the final bound decreases geometrically, and thus our bound does not diverge as increasing the time step $T$, and the gradient variance around final steps become dominant, which is consistent with the empirical study [Jiang et al., 2020].

---

> > > > ### Author Response · Authors · 2023-08-18
> > > > **Author responses (4)**
> > > >
> > > > ## Part (iii): Relevance to [SSS'08] and [L'22]
> > > >
> > > > ---
> > > > ### Part (iii)-1: Related to [SSS'08]
> > > >
> > > > As you pointed out, the obtained parameters via SGLD do not converge to the (local) minima under a fixed temperature parameter ($\beta$) for the Gaussian noise coefficient. These converge rather to a stationary distribution known as the Gibbs posterior $\pi(\mathrm{d}w) \propto e^{-\beta F(W, S)}\mathrm{d}w$ when $\beta$ is fixed over time step since SGLD is a ***sampling*** method from a target distribution based on the concept of Langevin dynamics, where $F(W, S)$ represents the training loss.
> > > > In essence, the trajectory of parameters via the SGLD algorithm gets closer to the minima and then explores its vicinity due to the addition of Gaussian noise to the gradient. Therefore, while ***convergence to a target distribution*** occurs, convergence to the minima itself is not achieved without controlling the noise via $\beta$.
> > > >
> > > > Although SGLD does not converge to the minima, as elucidated earlier, it boasts a distinct edge in its ability to explore parameters globally, even within non-convex problems, thanks to the Gaussian noise. This property enables the evaluation of how the obtained ***expected loss*** w.r.t. the stationary distribution deviates from that with the global minima. Specifically, we can evaluate this difference by factors that depend on parameter dimensions $d$, $\beta$, and the constants appearing in the assumptions for the potential function, such as dissipativity and smoothness, as elaborated in Appendix D.5 of our paper.
> > > > Furthermore, we can also derive the upper bounds for the population risk and excess risk both for convex and non-convex losses (please see [Raginsky et al., 2017; Xu et al., 2018]).
> > > >
> > > > ---
> > > > ### Part (iii)-2: Related to [LL'22]
> > > >
> > > > Here, we discuss the dependence on the parameter dimension $d$ pointed out in [L'22]. Unfortunately, removing this dependence is difficult or unavoidable even if our framework is utilized when analyzing the ***excess risk*** of discretized Langevin dynamics such as SGLD through the mutual information (MI) between the dataset and parameters.
> > > > This difficulty comes from the property that SGLD is the sampling method, as mentioned in Part (iii)-1. On the other hand, concerning the generalization error, existing IT-based bounds are expressed using gradient variance, explicitly independent of parameter dimension $d$ (we will mention this property in the specific example introduced in Part (iv)).
> > > > And the empirical fact that gradient variance decreases as learning progresses substantiates the significance and advantages of such algorithm- and data-dependent bounds, as we said in Part (ii).
> > > >
> > > > Despite the above merit, the gradient variance implicitly depends on the dimension of parameters $d$, as we mentioned in Part (ii) (it is known that the gradient variance gets small as training proceeds empirically, though). In order to theoretically mitigate this reliance on dimensionality, it could be imperative to explore an alternative approach to evaluating generalization that deviates from the mutual information (MI) between parameters and data, which forms the cornerstone of this paper.
> > > >
> > > > One possible avenue is, for instance, the utilization of conditional mutual information (CMI) involving super-samples (e.g., [Wang and Mao, 2023]), as highlighted in [L'22], as well as methods to quantify the MI between the ***learned hypothesis*** and dataset (hMI), instead of focusing on the parameters [Harutyunyan et al., 2021].
> > > > However, the drawback of these approaches is that it becomes challenging to explicitly incorporate statistics directly obtained from algorithms, such as the gradient variance, into the understanding and evaluation of generalization despite being analyses of algorithm-dependent generalization performance.
> > > > Seeking IT-based bounds that not only represent algorithm- and data-dependent statistics related to generalization performance, such as gradient variance but also theoretically eliminate dimension dependence constitutes a significant future work in the context of the IT-based generalization analysis field.

---

> > > > > ### Author Response · Authors · 2023-08-18
> > > > > **Author responses (5)**
> > > > >
> > > > > ## Part (iv) Specific examples demonstrating the effectiveness of our approach.
> > > > >
> > > > > In our analysis, as mentioned in Part (iii)-2, we upper bound the KL divergence by $\textrm{(KL-divergence at the previous time step)} \cdot e^{- \textrm{Const.}} + \textrm{(gradient variance at the previous time step)} + (\textrm{an additional term} c_3\eta$ (please refer to Eq. (7) in Lemma 1).
> > > > > If the third term is absent, the final KL divergence is dominantly expressed by the gradient variance at around the final time step.
> > > > > This is because the gradient variance at the early time-step geometrically diminishes due to the term $e^{- \textrm{Const.}}$ (please recall that this KL-divergence term is expressed recursively via one-step old gradient variance at each step).
> > > > >
> > > > > As for the third term in Eq. (7), this term explicitly depends on the algorithm-independent statistics, such as the smoothness parameter of the loss function and the dimension of the model. As a result, the inherent advantage of IT-based generalization error analysis, which allows for a concise explanation of generalization performance through gradient variance, is somewhat compromised (this is a limitation of our approach). On the other hand, we have found that in certain cases, such as ERM with convexity losses or with L2 regularization, we can eliminate this problematic third term ($c_3\eta$), i.e., bounds evaluated solely based on gradient variances. Furthermore, in this bound, the (initial) gradient variance from two or more time steps ago decreases geometrically, which is the ideal result we were aiming for. We explain the details below.
> > > > >
> > > > > For simplicity, we consider Langevin dynamics (LD) in the continuous limit instead of SGLD, which is the same setup as in Eq. (9) in Sec. 3.3 of our paper. As one particular case,  i) we assume that the training loss $F(w,z)$ is $R$-strongly convex ($0 < R< \infty$). We note that $R$-strongly convex losses satisfy dissipativity. As the other case, ii) we consider the ERM with $l_{2}$ regularization and the Gaussian initial distribution, i.e., $F(w,z)=F_0(w,z)+\frac{\lambda}{2}\||w\||^2$ ($0 < \lambda< \infty$), where the Gaussian initial distribution is given as $\pi_0\propto e^{-\beta\frac{\lambda}{2}\||w\||^2}$. We also assume that $F_0(w,z)$ is $C$-bounded ($0 \leq C < 0$). Note that we can use non-convex losses in this case if the above assumption holds because we add $l_{2}$ regularization so that $F(w,z)$ becomes a dissipative function.
> > > > >
> > > > > Under the above settings, we can derive the following inequality, which corresponds to Eq. (7) in Lemma 1. In the case i), when considering $R$-strongly convex ($0 < R< \infty$) loss and assume that $T=\eta K$ for $K\in \mathbb{N}$, we have
> > > > >
> > > > > $\mathrm{KL}(\rho_{\eta K}|\gamma_{\eta K}) \leq e^{-\frac{\eta}{2\beta R}}\mathrm{KL}(\rho_{\eta (K-1)}|\gamma_{\eta (K-1)})+\frac{\beta}{2} \int_{\eta (K-1)}^{\eta K}e^{-\frac{(\eta-t)}{2\beta R}} \mathbb{E}\|| \nabla F(W_t,S) - \nabla F(W_t,S')\||^{2}\mathrm{d}t$.
> > > > >
> > > > > In case ii), when considering $l_{2}$ regularization with $C$-bounded loss function, we obtain the same result as above by only replacing $R\to \frac{\lambda}{e^{8\beta C}}$.
> > > > > Unlike Eq. (7) in Lemma 1, which is shown under general dissipative loss functions, this new bound is expressed only in terms of the gradient variance, without the third term $c_3\eta$.
> > > > > Thanks to this result, only the gradient variance closer to the end time of the algorithm contributes to the bound values since the old gradient variances are geometrically reduced.
> > > > > This behavior is closer to the fact that past gradient variance, especially in the early time step, does not correlate well with the generalization performance of NNs [Jiang et al., 2020]. We also show the generalization error bound based on the above new bound;
> > > > >
> > > > > $|\mathrm{gen}(\mu,P_{W_T|S})|\leq \sqrt{\frac{\beta \sigma_g^2}{n}\sum_{k=0}^{K-1}\int_{\eta k}^{\eta(k+1)}e^{-\frac{(\eta (K-k)-t)}{2\beta R}} \mathbb{E}\|| \nabla F(W_t,S)- \nabla F(W_t,S')\||^{2}\mathrm{d}t}$.
> > > > >
> > > > > The difference from Theorem 4 in our paper is that the bound can be expressed only in terms of the gradient variance at each time step. We note that the new bound does not lose the property of time independence and geometrical reduction of the gradient variance at early time steps.
> > > > > *(Continue to the next thread)*

---

> > > > > > ### Author Response · Authors · 2023-08-18
> > > > > > **Author responses (6)**
> > > > > >
> > > > > > *(Continued from previous thread)*
> > > > > > In the case of SGLD (LD with discretization), a similar bound can be derived using conditional variance, as shown in Appendix C-1. This result cannot be obtained simply by applying data processing inequality to the MI between the dataset and parameters evaluated in the existing Information-theoretic bounds. This is the non-trivial result derived by our approach based on the geometric information that SGLD is a Markov diffusion process given by a stochastic differential equation called LD and that the amount of Fisher information at each time point is bounded by the KL-divergence. Also, this bound is closely related to Proposition 9 of [Mou et al., 18], which evaluated the generalization in terms of the ***gradient norm***. The bound of that paper evaluates the generalization error bound through the norm of the gradient, whereas our bound is expressed by the gradient variance, a statistic that is related to the stability of the algorithm.
> > > > > > As a conclusion to this part, we provide the facts we used in proving this bound.
> > > > > > In the case of strongly convex losses, we utilize local Logarithmic Sobolev inequality from [Bakry, 2013]; in the case of bounded losses + L2 regularization, we use Lemma 34 in [Li et al., 2020]. The complete proof will be organized in Appendix E.
> > > > > >
> > > > > > ---
> > > > > > ## Acknowledgement & Conclusion
> > > > > >
> > > > > > We genuinely appreciate your insightful remarks and the opportunity to consider specific examples where our approach is also valid in the context of convex analysis.
> > > > > > It's truly gratifying to know that we could come up with such interesting examples as mentioned above. Additionally, I'm grateful for your acknowledgment of the concerns related to the dependence on parameter dimensions and convergence of optimization.
> > > > > > As pointed out not only in the paper by [L’22] but also in [Haghifam et al., 2023], limitations of information-theoretic generalization bounds for gradient descent methods in stochastic convex optimization], this is an issue that pertains to the entire community of generalization analysis based on information-theoretic approaches, including ours.
> > > > > > Ensuring consistency with convergence properties is indeed a crucial direction for the future, without a doubt.
> > > > > >
> > > > > > The bounds introduced in Part (iii) seem to provide important and intriguing insights, and we believe they should be added to the main part of our paper. Therefore, we will create a new subsection (Section 3.4) and explain the above.
> > > > > > Furthermore, discussions regarding convergence properties and dimension dependence will be included and the new appendix section (Appendix E.: Additional discussion of limitations and related studies in our study)
> > > > > >
> > > > > > ---
> > > > > > ## New References
> > > > > > - [Jiang et al., 2020]: Y. Jiang, B. Neyshabur, H. Mobahi, D. Krishnan, and S. Bengio. ``Fantastic Generalization Measures and Where to Find Them.’’ In The Eighth International Conference on Learning Representations, 2020.
> > > > > > https://openreview.net/pdf?id=SJgIPJBFvH
> > > > > >
> > > > > > - [Amit et al., 2022]: R. Amit, B. Epstein, S. Moran, and R. Meir. ``Integral Probability Metrics PAC-Bayes Bounds.’’ In Advances in Neural Information Processing Systems, vol. 35, pp. 3123–3136, 2022.
> > > > > > https://openreview.net/pdf?id=H547BtAyOJ4
> > > > > >
> > > > > > - [Wang and Mao, 2023]: Z. Wang and Y. Mao. ``Tighter information-theoretic generalization bounds from supersamples.’’ In Proceedings of the 40th International Conference on Machine Learning, 2023.
> > > > > > https://openreview.net/pdf?id=y6gg68aGiq
> > > > > >
> > > > > > - [Harutyunyan et al., 2021]: H. Harutyunyan, M. Raginsky, G. Ver Steeg, and A. Galstyan. ``Information-theoretic generalization bounds for black-box learning algorithms.’’ In Advances in Neural Information Processing Systems, vol. 34, pp.24670–24682, 2021.
> > > > > > https://openreview.net/pdf?id=L_cN8vD0XdT
> > > > > >
> > > > > > - [Mou et al., 18]: W. Mou, L. Wang, X. Zhai, and K. Zheng. ``Generalization bounds of SGLD for non-convex learning: Two theoretical viewpoints.’’ In Proceedings of the 31st Conference On Learning Theory, volume 75, pages 605–638, 2018.
> > > > > > http://proceedings.mlr.press/v75/mou18a/mou18a.pdf
> > > > > >
> > > > > > - [Li et ai., 2020]: J. Li, X. Luo, and M. Qiao. ``On Generalization Error Bounds of Noisy Gradient Methods for Non-Convex Learning.’’ In The Eighth International Conference on Learning Representations, 2020.
> > > > > > https://openreview.net/pdf?id=SkxxtgHKPS
> > > > > >
> > > > > > - [Haghifam et al., 2023]: M. Haghifam, B. Rodríguez-Gálvez, R. Thobaben, M. Skoglund, D. M. Roy, and G. K. Dziugaite. ``Limitations of information-theoretic generalization bounds for gradient descent methods in stochastic convex optimization.’’ In Proceedings of the 34th Conference On Learning Theory, volume 201, pages 663–706, 2023.
> > > > > > https://proceedings.mlr.press/v201/haghifam23a/haghifam23a.pdf

---

### Decision · Program_Chairs · 2023-09-21

**Decision:**

Accept (poster)

**Comment:**

- The paper provides a new time-independent information theoretic generalization bound.

- The bound was found to be interesting by the reviewers who favored the paper.

- During discussion, a limitation was raised, that no proof-of-concept was proposed for the utility of the bound.
- It was mentioned, that potentially the bound might be relevant only when there is some dependence on the dimension -- in that case time-independent bounds are trivial (uniform convergence)
- The authors raised the fact that this is indeed a limitation but may be a limitation to the whole approach of information-theoretic bounds -- they are correct in this.
- This limitation should be further discussed in the paper, and hopefully the authors could present a proof of concept for a case where the information bound is useful.

- In the context of this paper, please note that L'22 provides a dimension-dependent lower bound on the mutual information that relies on a dissipative function (in fact a function of the form $\|w\|^2 + F_0(w)$. In other words, the structural assumptions suggested by the authors cannot circumvent dimension dependence without further assumptions.